# END-TO-END LEARNING OF GAUSSIAN MIXTURE PRIORS FOR DIFFUSION SAMPLER

**Denis Blessing**[*,1], **Xiaogang Jia**[1], **Gerhard Neumann**[1,2]
[1]Autonomous Learning Robots, Karlsruhe Institute of Technology
[2]FZI Research Center for Information Technology

## ABSTRACT

Diffusion models optimized via variational inference (VI) have emerged as a promising tool for generating samples from unnormalized target densities. These models create samples by simulating a stochastic differential equation, starting from a simple, tractable prior, typically a Gaussian distribution. However, when the support of this prior differs greatly from that of the target distribution, diffusion models often struggle to explore effectively or suffer from large discretization errors. Moreover, learning the prior distribution can lead to mode-collapse, exacerbated by the mode-seeking nature of reverse Kullback-Leibler divergence commonly used in VI. To address these challenges, we propose end-to-end learnable Gaussian mixture priors (GMPs). GMPs offer improved control over exploration, adaptability to target support, and increased expressiveness to counteract mode collapse. We further leverage the structure of mixture models by proposing a strategy to iteratively refine the model by adding mixture components during training. Our experimental results demonstrate significant performance improvements across a diverse range of real-world and synthetic benchmark problems when using GMPs without requiring additional target evaluations.

## 1 INTRODUCTION

Sampling methods are designed to address the challenge of generating approximate samples or estimating the intractable normalization constant $\mathcal{Z}$ for a probability density $\pi$ on $\mathbb{R}^d$ of the form

$$\pi(x) = \frac{\rho(x)}{\mathcal{Z}}, \quad \mathcal{Z} = \int_{\mathbb{R}^d} \rho(x)\mathrm{d}x, \tag{1}$$

where $\rho : \mathbb{R}^d \to (0, \infty)$ can be evaluated. This formulation has broad applications in fields such as Bayesian statistics, the natural sciences (Liu & Liu, 2001; Stoltz et al., 2010; Frenkel & Smit, 2023; Schopmans & Friederich, 2024), or robotics (Zhou et al., 2024; Celik et al., 2025).

Monte Carlo (MC) methods (Hammersley, 2013), Annealed Importance Sampling (AIS) (Neal, 2001), and their Sequential Monte Carlo (SMC) extensions (Del Moral et al., 2006; Arbel et al., 2021; Matthews et al., 2022; Midgley et al., 2022) have long been regarded as the gold standard for tackling complex sampling problems. An alternative approach is variational inference (VI) (Blei et al., 2017), which approximates an intractable target distribution by parameterizing a family of tractable distributions. Recently, there has been growing interest in diffusion models (Zhang & Chen, 2021; Berner et al., 2022; Richter et al., 2023; Vargas et al., 2023a;b; Grenioux et al., 2024), which employ stochastic processes to transport samples from a simple, tractable prior distribution to the target distribution. While diffusion models have shown great success in generative modeling (Ho et al., 2020; Song et al., 2020), their application to sampling tasks introduces unique challenges.

We identify these challenges as follows (**C1–C3**): Unlike generative modeling, where the support of the target distribution is often known, in sampling tasks, the target's support is usually unknown. This makes it difficult to set the prior appropriately and requires the model to explore the relevant regions of the space—an exploration that becomes exponentially harder as dimensionality increases (**C1**). Additionally, large discrepancies between the support of the prior and the target distribution

---

[*]Correspondence to `denis.blessing@kit.edu`

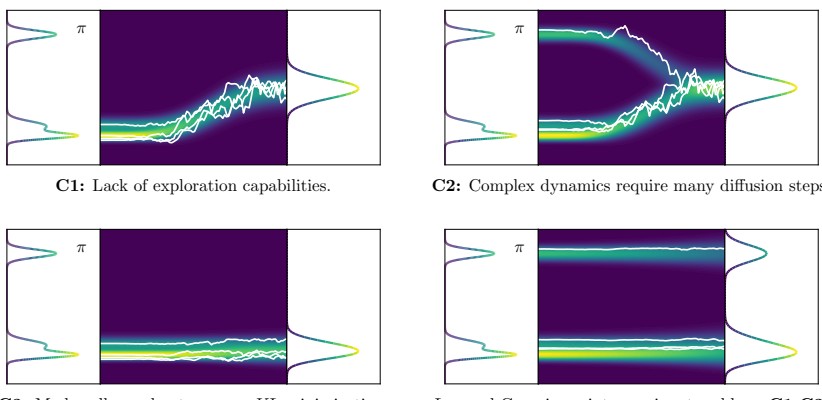

C1: Lack of exploration capabilities.

C2: Complex dynamics require many diffusion steps.

C3: Mode collapse due to reverse KL minimization.

Learned Gaussian mixture priors to address C1-C3.

Figure 1: Illustration of challenges (**C1-C3**) associated with diffusion-based sampling methods and how learned Gaussian mixture priors address them (bottom right). Here, $\pi$ denotes the target distribution.

can lead to highly non-linear dynamics, necessitating many diffusion steps to mitigate discretization errors (**C2**). Finally, while joint optimization of the prior and diffusion process is possible, using simple priors like Gaussians can result in mode collapse due to the mode-seeking behavior of the reverse Kullback-Leibler (KL) divergence commonly used in VI (**C3**). These challenges are illustrated in Figure 1.

**Outline.** In Section 3, we present an overview of diffusion-based sampling methods within the framework of variational inference. Next, we discuss the necessary adaptations for supporting the learning of arbitrary prior distributions, illustrated through specific examples of diffusion models (Section 4). We then provide a rationale for our choice of Gaussian Mixture Priors (GMPs) and introduce a novel training scheme designed to iteratively refine diffusion models during training (Section 5). Finally, in Section 6, we assess our method through experiments on a range of real-world and synthetic benchmark problems, demonstrating consistent improvements in performance.

## 2 RELATED WORK

**Sampling and Variational Inference.** Numerous works have studied the problem of sampling from unnormalized densities to estimate the partition function $Z$, including Monte Carlo (MC) methods such as Markov Chain Monte Carlo (MCMC) (Kass et al., 1998) and Sequential Importance Sampling (Liu et al., 2001). Seminal works include Annealed Importance Sampling (Neal, 2001; Guo et al., 2025) and its Sequential Monte Carlo extensions (Del Moral et al., 2006; Arbel et al., 2021; Wu et al., 2020; Matthews et al., 2022; Midgley et al., 2022). Another line of work approaches the sampling problem by utilizing tools from optimization to fit a parametric family of distributions to the target density $\pi$, known as Variational Inference (VI) (Blei et al., 2017). To that end, one typically uses the reverse Kullback-Leibler divergence, although other discrepancies have been studied (Li & Turner, 2016; Midgley et al., 2022; Dieng et al., 2017; Richter et al., 2020; Wan et al., 2020; Naesseth et al., 2020).

**Diffusion-based Sampling Methods.** Recently, there has been growing interest in combining Monte Carlo methods with variational techniques by constructing a sequence of variational distributions through the parameterization of Markov chains (Naesseth et al., 2018; Geffner & Domke, 2021; Thin et al., 2021; Zhang et al., 2021; Chen et al., 2024b). In the limit of infinitely many steps, these Markov chains converge to stochastic differential equations (SDEs) (Särkkä & Solin, 2019), which has led to further research on diffusion-based models for sampling, particularly in light of advances in generative modeling (Ho et al., 2020; Song et al., 2020). One line of work considered parameterized drift functions to improve annealed Langevin diffusions in the overdamped (Doucet et al., 2022a) or underdamped (Geffner & Domke, 2022) regime. Another line of work casts diffusion-based sampling as a stochastic optimal control problem (Dai Pra, 1991) including denoising diffusion models (Berner et al., 2022; Vargas et al., 2023a), and Follmer sampling (Föllmer, 2005; Zhang & Chen, 2021; Vargas et al., 2023b). A unifying view was later provided by Vargas et al. (2024); Richter et al. (2023); Blessing et al. (2025). Further extensions to diffusion-based sampling methods have been proposed such as improved learning objectives (Zhang et al., 2023;

Akhound-Sadegh et al., 2024; Noble et al., 2024) or combinations with sequential importance sampling (Phillips et al., 2024; Chen et al., 2024a). Another study leverages physics-informed neural networks (PINNs, (Raissi et al., 2019)) to learn the Fokker-Planck equation governing the density evolution of the diffusion process (Sun et al., 2024; Shi et al., 2024).

## 3 PRELIMINARIES

In this section, we offer a concise overview of diffusion models within the context of variational inference. Our discussion draws primarily from the works of Richter et al. (2023); Vargas et al. (2024). While these studies emphasize the continuous-time perspective, we adopt an approach that largely emphasizes discrete time, aiming to make this work more accessible to readers without a background in stochastic calculus.

### 3.1 CONTROLLED DIFFUSIONS, DISCRETIZATION, AND COUPLINGS

We consider two $\mathbb{R}^d$-valued stochastic processes $(X_t)_{t\in[0,T]}$ on the time-interval $[0,T]$: One starts from a prior distribution $p_0$ and runs forward in time whereas the other starts from the target distribution $p_T = \pi$ and runs backward in time. These processes are governed by the stochastic differential equations (SDEs) given by controlled diffusions, that is,

$$\mathrm{d}X_t = \left[f(X_t,t) + \sigma u^\theta(X_t,t)\right]\mathrm{d}t + \sqrt{2}\sigma\mathrm{d}B_t, \quad X_0 \sim p_0, \tag{2a}$$

$$\mathrm{d}X_t = \left[f(X_t,t) - \sigma v^\gamma(X_t,t)\right]\mathrm{d}t + \sqrt{2}\sigma\mathrm{d}B_t, \quad X_T \sim p_T = \pi, \tag{2b}$$

with drift, and parameterized control functions $f, u^\theta, v^\gamma : \mathbb{R}^d \times [0,T] \to \mathbb{R}^d$, respectively. Further, $(B_t)_{t\in[0,T]}$ is a $d$-dimensional Brownian motion and $\sigma \in \mathbb{R}^+$ a diffusion coefficient. For integration, we consider the Euler-Maruyama (EM) method with constant discretization step size $\delta t \geq 0$ such that $N = T/\delta t$ is an integer. To simplify notation, we write $x_n$, instead of $x_{n\delta t}$. Integrating (2a) yields

$$x_{n+1} = x_n + \left[f(x_n,n) + \sigma u^\theta(x_n,n)\right]\delta t + \sigma\sqrt{2\delta t}\epsilon_n, \quad x_0 \sim p_0, \tag{3}$$

where $\epsilon_n \sim \mathcal{N}(0,I)$. The EM discretizations of (2a) and (2b) admit the following Markov Processes

$$\vec{p}^{\,\theta}(x_{0:N}) = p_0(x_0)\prod_{n=1}^{N}\vec{p}^{\,\theta}_{n|n-1}(x_n|x_{n-1}), \quad \text{and} \tag{4}$$

$$\reflectbox{$\vec{p}$}^{\,\gamma}(x_{0:N}) = p_T(x_N)\prod_{n=1}^{N}\reflectbox{$\vec{p}$}^{\,\gamma}_{n-1|n}(x_{n-1}|x_n), \tag{5}$$

in a sense that $\vec{p}^{\,\theta}$ and $\reflectbox{$\vec{p}$}^{\,\gamma}$ converge to their law, respectively, as $\delta t \to 0$. Here,

$$\vec{p}^{\,\theta}_{n+1|n}(x_{n+1}|x_n) = \mathcal{N}\left(x_{n+1}|x_n + \left[f(x_n,n) + \sigma u^\theta(x_n,n)\right]\delta t, 2\sigma^2\delta tI\right), \quad \text{and} \tag{6}$$

$$\reflectbox{$\vec{p}$}^{\,\gamma}_{n-1|n}(x_{n-1}|x_n) = \mathcal{N}\left(x_{n-1}|x_n - \left[f(x_n,n) - \sigma v^\gamma(x_n,n)\right]\delta t, 2\sigma^2\delta tI\right). \tag{7}$$

The goal of diffusion-based sampling methods is to align these processes by learning control functions $u^\theta$ and $v^\gamma$ such that

$$\vec{p}^{\,\theta}(x_{0:N}) = \reflectbox{$\vec{p}$}^{\,\gamma}(x_{0:N}). \tag{8}$$

Assuming (8) holds, we have $\int \vec{p}^{\,\theta}(x_{0:N})\mathrm{d}x_{0:N-1} = \int \reflectbox{$\vec{p}$}^{\,\gamma}(x_{0:N})\mathrm{d}x_{0:N-1} = \pi(x_N)$, meaning that we can sample $x_0 \sim p_0$ and integrate the 'forward' diffusion process (2a) to obtain samples from $\pi$. In contrast, the 'backward' process (2b) is not needed for generating samples from $\pi$, but is required for estimating the normalization constant $\mathcal{Z}$, and obtaining a tractable optimization objective which is discussed in the next section. Lastly, we want to highlight that this formulation of diffusion-based sampling is very generic and that most instances of samplers, such as denoising diffusion samplers (Berner et al., 2022; Vargas et al., 2023a), can be recovered by choosing the drift and control functions in (2) appropriately. We refer the interested reader to Richter et al. (2023) for further details.

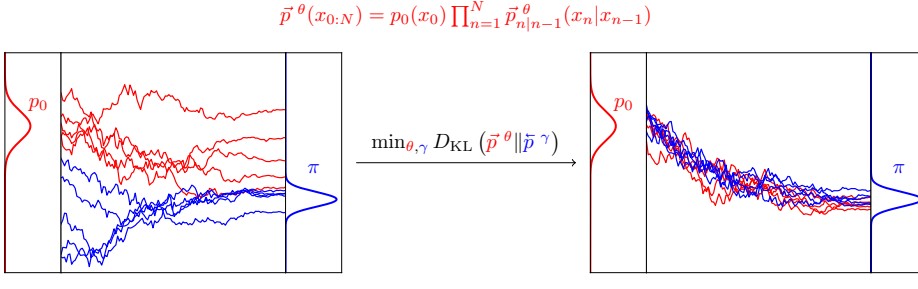

$$\vec{p}^{\,\theta}(x_{0:N}) = p_0(x_0) \prod_{n=1}^{N} \vec{p}^{\,\theta}_{n|n-1}(x_n|x_{n-1})$$

$$\breve{p}^{\,\gamma}(x_{0:N}) = \pi(x_N) \prod_{n=1}^{N} \breve{p}^{\,\gamma}_{n-1|n}(x_{n-1}|x_n)$$

Figure 2: Diffusion-Based Sampling: The goal is to align two parameterized Markov Processes $\vec{p}^{\,\theta}$ and $\breve{p}^{\,\gamma}$. The former starts at the prior $p_0$ and runs forward in time while the latter starts at the target $\pi$ and runs backward.

## 3.2 Variational Inference for Diffusion Models

Variational Inference (Blei et al., 2017) uses a parameterized tractable distribution $p^\theta$ and minimizes a divergence to the target distribution $\pi$ with respect to its parameters $\theta$, typically the Kullback-Leibler divergence, i.e.,

$$D_{\mathrm{KL}}\left(p^\theta(x)\|\pi(x)\right) = \mathbb{E}_{x\sim p^\theta}\left[\log\frac{p^\theta(x)}{\rho(x)}\right] + \log\mathcal{Z} = -\mathrm{ELBO}(\theta) + \log\mathcal{Z}, \qquad (9)$$

It directly follows that minimizing $D_{\mathrm{KL}}$, or equivalently, maximizing the ELBO[1] does not require access to the true normalization constant $\mathcal{Z}$ as it is independent of $\theta$. Moreover, using the fact that $D_{\mathrm{KL}} \geq 0$, it is straightforward to see that $\mathrm{ELBO}(\theta) \leq \log\mathcal{Z}$.

In the case of diffusion models, we are interested in learning the parameters $\theta, \gamma$ of the control functions $u^\theta, v^\gamma$. Directly minimizing $D_{\mathrm{KL}}$ between $p^\theta_T(x_N) = \int \vec{p}^{\,\theta}(x_{0:N})\mathrm{d}x_{0:N-1}$ and $\pi$ is challenging. However, the data-processing inequality (Cover, 1999)

$$D_{\mathrm{KL}}\left(p^\theta_T(x_T)\|\pi(x_T)\right) \leq D_{\mathrm{KL}}\left(\vec{p}^{\,\theta}(x_{0:N})\|\breve{p}^{\,\gamma}(x_{0:N})\right), \qquad (10)$$

provides an auxiliary, tractable, objective for optimizing $(\theta, \gamma)$, that is,

$$D_{\mathrm{KL}}\left(\vec{p}^{\,\theta}(x_{0:N})\|\breve{p}^{\,\gamma}(x_{0:N})\right) = \underbrace{\mathbb{E}_{x_{0:N}\sim\vec{p}^{\,\theta}}\left[\log\frac{p_0(x_0)}{\rho(x_N)} + \sum_{n=1}^{N}\log\frac{\vec{p}^{\,\theta}_{n|n-1}(x_n|x_{n-1})}{\breve{p}^{\,\gamma}_{n-1|n}(x_{n-1}|x_n)}\right]}_{-\mathcal{L}(\theta,\gamma)} + \log\mathcal{Z},$$

$$(11)$$

where $\mathcal{L}(\theta, \gamma)$ is often referred to as augmented or extended evidence lower bound, as it has additional looseness due to the latent variables $x_{0:N-1}$ (Geffner & Domke, 2021). Note that the VI setting for optimizing diffusion models is different from techniques used when samples from the target, i.e., $x_N \sim \pi$ are available. The former requires simulations $x_{0:N} \sim \vec{p}^{\,\theta}$ for optimization, while the latter minimizes the forward KL $D_{\mathrm{KL}}\left(\breve{p}^{\,\gamma}\|\vec{p}^{\,\theta}\right)$, allowing for simulation-free optimization techniques such as denoising score-matching (Vincent, 2011; Song & Ermon, 2019) or bridge matching (Liu et al., 2022; Shi et al., 2024). Moreover, recent works consider minimizing other loss functions that the KL divergence in (11). A recent overview of possible alternatives can be found in Domingo-Enrich (2024). For further details, the interested reader is referred to Berner et al. (2022); Vargas et al. (2024).

## 4 End-to-End Learning of Prior Distributions

We aim to learn a parametric prior $p_0^\phi$ with parameters $\phi$ end-to-end when maximizing the extended ELBO $\mathcal{L}$ ( (11)). To that end, we consider two requirements:

1. We can compute gradients of $\mathcal{L}$ with respect to $\phi$.

---

[1]Evidence Lower Bound. The terminology stems from Bayesian inference, where $\log\mathcal{Z}$ is equivalent to the evidence of the data.

2. There exists a $\phi$ and $\gamma$ such that $p_0^\phi(x_0) = \int \overset{\leftarrow}{p}{}^\gamma(x_{0:N})\mathrm{d}x_{1:N}$.

For the former, we assume that $p_0^\phi(x_0)$ is amendable to the reparameterization trick[2], i.e., we can express a sample $x_0$ from $p_0^\phi$ as a deterministic function of a random variable $\xi$ with some fixed distribution and the parameters $\phi$, i.e., $x_0 = g(\xi, \phi)$. We can then obtain gradients of

$$\mathcal{L}(\theta, \gamma, \phi) = \mathbb{E}_{x_{0:N} \sim \vec{p}^{\,\theta,\phi}}\left[\log \frac{\rho(x_N)}{p_0^\phi(x_0)} + \sum_{n=1}^{N} \log \frac{\overset{\leftarrow}{p}{}_{n-1|n}^\gamma(x_{n-1}|x_n)}{\vec{p}_{n|n-1}^\theta(x_n|x_{n-1})}\right], \tag{12}$$

with $\vec{p}^{\,\theta,\phi}(x_{0:N}) = p_0^\phi(x_0)\prod_{n=1}^{N}\vec{p}_{n|n-1}^\theta(x_n|x_{n-1})$, with respect to $\phi$, by differentiating through the stochastic process

$$x_{n+1} = x_n + \left[f(x_n, n) + \sigma u^\theta(x_n, n)\right]\delta t + \sigma\sqrt{2\delta t}\epsilon_n, \quad x_0 = g(\xi, \phi). \tag{13}$$

The second requirement is necessary to obtain a coupling between $p_0^\phi$ and $\pi$, i.e., to satisfy (8). This requirement is trivially fulfilled for a controlled backward process (2b), where we can learn a $v^\gamma$ such that $\overset{\leftarrow}{p}{}^\gamma$ transports $\pi$ back to $p_0^\phi$. However, this requirement can be more intricate for other processes and will be discussed in the next sections. In particular, we look at two instances of (2), namely denoising diffusion models (Berner et al., 2022; Vargas et al., 2023a) and annealed Langevin diffusions (Doucet et al., 2022a; Vargas et al., 2024).

## 4.1 DENOISING DIFFUSION MODELS

Denoising diffusion models use an Ornstein Uhlenbeck (OU) process for (2b), that is [3],

$$\mathrm{d}X_t = \sigma^2 X_t\mathrm{d}t + \sqrt{2}\sigma\mathrm{d}B_t, \quad X_T \sim p_T = \pi, \tag{14}$$

and, hence, a special case of (2) with $f(X_t, t) = \sigma^2 X_t$ and $v^\gamma = 0$. Assuming a sufficiently large $\sigma$ (or $T$), it holds that $p_0(x_0) = \int \overset{\leftarrow}{p}(x_{0:N})\mathrm{d}x_{1:N} \approx \mathcal{N}(0, I)$. In other words, the OU process transports the target $\pi$ to a Gaussian distribution. We extend denoising diffusion models to support learning arbitrary priors based on Proposition 1, whose proof can be found in Appendix A.1.

**Proposition 1.** *Let (2b) be a (uncontrolled) stochastic process as defined in (2) with $v^\gamma = 0$, starting from $p_T = \pi$. For a time-independent drift, i.e., $f(x, t) = f(x)$, the stationary distribution $p^{st}(x)$ for which $\frac{\partial p_t(x_t)}{\partial t} = 0$ holds, is given by*

$$p^{st}(x) = \frac{1}{Z^{st}}\exp\left(-\frac{1}{\sigma^2}\int f(x)\mathrm{d}x\right), \tag{15}$$

*with normalization constant $Z^{st}$.*

Rewriting (15), yields $f = -\sigma^2\nabla_x\log p^{st}$, resulting in the SDE

$$\mathrm{d}X_t = -\sigma^2\nabla_x\log p^{st}(X_t)\mathrm{d}t + \sqrt{2}\sigma\mathrm{d}B_t, \quad X_T \sim p_T = \pi, \tag{16}$$

with stationary distribution $p^{st}(x)$. Note that denoising diffusion models leverage this result by setting $p^{st} = \mathcal{N}(0, I)$, resulting in the OU process ((14)) since $\nabla_x\log p^{st}(x) = -x$. Hence, we can adapt existing denoising diffusion sampling methods (Vargas et al., 2023a; Berner et al., 2022) to arbitrary priors $p^\phi$ using

$$\mathrm{d}X_t = -\sigma^2\nabla_x\log p^\phi(X_t)\mathrm{d}t + \sqrt{2}\sigma\mathrm{d}B_t, \quad X_T \sim p_T = \pi. \tag{17}$$

However, contrary to the OU process, where the relaxation time, i.e., the time scale over which the system loses memory of its initial conditions and approaches its stationary distribution, can be estimated analytically, it is unknown for general $p^\phi$ and is only guaranteed as $T \to \infty$ (Roberts & Tweedie, 1996). We address this by additionally learning the time horizon $T = N\delta t$ by treating the discretization step size $\delta t$ as a learnable parameter. As such, the parameters $\phi$ of the stationary

---

[2]Note that this requirement is not necessary when minimizing loss function where the expectation is not computed with respect to samples from $\vec{p}^{\,\theta}$. For further details see e.g. (Richter et al., 2023).

[3]The sign differs here compared to most works as the SDE is integrated backward in time.

distribution, i.e., the prior distribution and the discretization step size $\delta t$ are optimized jointly by maximizing the extended ELBO

$$\mathcal{L}(\theta, \phi) = \mathbb{E}_{x_{0:N} \sim \vec{p}^{\,\theta,\phi}} \left[ \log \frac{\rho(x_N)}{p_0^\phi(x_0)} + \sum_{n=1}^{N} \log \frac{\overset{\leftarrow}{p}_{n-1|n}^{\phi}(x_{n-1}|x_n)}{\vec{p}_{n|n-1}^{\,\theta,\phi}(x_n|x_{n-1})} \right], \tag{18}$$

with additional parameters $\phi, \delta t$. Please note that we omit dependence on $\delta t$ in Equation (18) to keep the notation uncluttered. Proposition 1 thus suggests, that for any $\phi$, there exists a $\delta t$ such that $p_0^\phi(x_0) = \int \overset{\leftarrow}{p}^{\,\phi}(x_{0:N}) \mathrm{d}x_{1:N}$ as $N \to \infty$. Empirically, we observe substantial improvements for finite values of $N$, as demonstrated in Section 6

## 4.2 ANNEALED LANGEVIN DIFFUSIONS

Annealed Langevin Diffusions use an annealed version of the (overdamped) Langevin diffusion equation by constructing a sequence of distributions $(\pi_t)_{t \in [0,T]}$ that anneal smoothly from the prior distribution $\pi_0 = p_0$ to the target distribution $\pi_T = \pi$. One typically uses the geometric average, that is, $\pi_t(x) = p_0(x)^{\beta_t} \pi(x)^{1-\beta_t}$, for $\beta_t$ monotonically increasing in $t$ with $\beta_0 = 0$ and $\beta_T = 1$. When learning the prior, we can use a parametric annealing, i.e., $\pi_t^\phi(x) = p_0^\phi(x)^{\beta_t} \pi(x)^{1-\beta_t}$. The corresponding stochastic processes can be described as an instance of (2) given by

$$\mathrm{d}X_t = \left[ \sigma^2 \nabla_x \log \pi_t^\phi(X_t) + \sigma u^\theta(X_t, t) \right] \mathrm{d}t + \sqrt{2}\sigma \mathrm{d}B_t, \quad X_0 \sim p_0 = p^\phi, \tag{19}$$

$$\mathrm{d}X_t = \left[ \sigma^2 \nabla_x \log \pi_t^\phi(X_t) - \sigma v^\gamma(X_t, t) \right] \mathrm{d}t + \sqrt{2}\sigma \mathrm{d}B_t, \quad X_T \sim p_T = \pi, \tag{20}$$

when setting $f = \nabla_x \log \pi_t^\phi$. Note that $\nabla_x \log \pi_t^\phi$ can be computed without knowing the normalization constant $\mathcal{Z}$ of $\pi$. Different variants can be derived from using either controlled or uncontrolled processes: Monte Carlo Diffusions (MCD) (Doucet et al., 2022b) uses a controlled process (2b) but uncontrolled (2a) and Controlled Monte Carlo Diffusions (CMCD) (Vargas et al., 2024) control both processes. Since both methods use controlled backward processes (2b), the second requirement is satisfied. Finally, while this work focuses on overdamped approaches, we want to highlight that there exist methods that are based on the underdamped Langevin equation (Geffner & Domke, 2021; Geffner & Domke, 2022), however, the idea of learning a prior end-to-end straightforwardly transfers to these approaches.

## 5 GAUSSIAN MIXTURE PRIORS AND ITERATIVE MODEL REFINEMENT

In this work, we focus on end-to-end learned Gaussian mixture priors (GMPs), that is,

$$p_0^\phi(x_0) = \sum_{k=1}^{K} \alpha_k p_0^{\phi_k}(x_0) = \sum_{k=1}^{K} \alpha_k \mathcal{N}(x_0|\mu_k, \Sigma_k), \quad \alpha_k \geq 0, \quad \sum_{k=1}^{K} \alpha_k = 1, \tag{21}$$

with mixture weights $\alpha_k$, Gaussian components $p_0^{\phi_k}(x_0) = \mathcal{N}(x_0|\mu_k, \Sigma_k)$ and parameters $\phi = \bigcup_{k=1}^{K} \{\alpha_k, \phi_k\}$ with $\phi_k = \{\mu_k, \Sigma_k\}$. Having established how the prior is learned in Section 4, we discuss desirable properties to address the challenges outlined in Section 1 and how GMPs address them.

A key objective is to improve the exploration capabilities of diffusion-based sampling methods to address **C1**. GMPs allow control over exploration by adjusting the initial variance of each Gaussian component. Additionally, the means of the Gaussian components can be initialized to incorporate prior knowledge of the target density, even if this knowledge is limited to a rough estimate of the target's support. This aspect will be elaborated on later in this section.

Another important consideration is to adjust the support of the prior such that it matches the target density, which reduces the complexity of the dynamics and, in turn, minimizes the number of diffusion steps required. GMPs demonstrate rapid adaptation capabilities, partially through their small parameter count, making them particularly suitable for addressing **C2**.

To prevent the model from focusing only on a subset of the target support (**C3**), which may occur due to the optimization of the mode-seeking reverse KL divergence, we require a more expressive

distribution than a single Gaussian prior. GMPs provide a solution by combining multiple Gaussian components, each of which can focus on different subsets of the target support.

Finally, efficient evaluation of $p_0^\phi$ is crucial, as it must be performed at each discretization step of the stochastic differential equation (SDE) that governs the diffusion process. This requirement is satisfied by GMPs, particularly when using diagonal covariance matrices.

**Iterative Model Refinement.** Gradually increasing the model complexity during the optimization process has demonstrated promising results in previous studies (Guo et al., 2016; Miller et al., 2017; Arenz et al., 2018; Cranko & Nock, 2019), and is directly applicable to our approach. We begin with an initial prior distribution $p_0^\phi = p_0^{\phi_1}$ with parameters $\phi_1$. These parameters are optimized using (18). After a predefined criterion is met, such as a fixed number of iterations, a second distribution $p_0^{\phi_2}$ is added, forming a new prior: $p_0^\phi = \alpha_1 p_0^{\phi_1} + \alpha_2 p_0^{\phi_2}$, with $\alpha \in \mathbb{R}^+$ and $\alpha_1 + \alpha_2 = 1$. This process is repeated, resulting in a mixture model $p_0^\phi(x) = \sum_{k=1}^K \alpha_k p_0^{\phi_k}(x)$.

We identify the benefits of this iterative scheme as twofold: First, it can simplify optimization by focusing on learning a subset of parameters $\phi_k$ at a time, rather than jointly optimizing all $\phi_k$ (Bengio et al., 2009). Second, it enables the initialization of newly added components based on a partially trained model, potentially preventing mixture components to focus on similar parts of the target support. For GMPs, for instance, the mean of a new component $\mu_{\text{new}}$ can be placed in a promising region, potentially informed by prior knowledge of the task or by running a $\pi$-invariant Markov chain to obtain a set of promising samples. More generally, consider a set of candidate samples $\mathcal{C} = \{x_i\}_{i=1}^C$. We propose initializing the mean of a new component $\mu_{\text{new}}$ as follows:

$$\mu_{\text{new}} = \arg\max_{x_N \in \mathcal{C}} \mathbb{E}_{x_{0:N-1} \sim \overleftarrow{p}^{\gamma,\phi}} \left[ \log \frac{\rho(x_N)}{p_0^\phi(x_0)} + \sum_{n=1}^N \log \frac{\overleftarrow{p}_{n-1|n}^{\gamma,\phi}(x_{n-1}|x_n)}{\overrightarrow{p}_{n|n-1}^{\theta,\phi}(x_n|x_{n-1})} \right], \tag{22}$$

where $p_0^\phi$ is the current model. This heuristic balances exploration and exploitation by favoring samples with high target likelihood and low prior likelihood, while also accounting for the diffusion process. See Appendix B for further details.

## 6 NUMERICAL EVALUATION

In this section, we test the impact of our proposed end-to-end learning scheme for prior distributions. Specifically, we consider three distinct settings: First, we evaluate these methods with a Gaussian prior that is fixed during training. Second and third, we consider learned Gaussian (GP) and Gaussian mixture priors (GMP). We indicate these different settings as X, X-GP, and X-GMP, respectively, where X is the corresponding acronym of the diffusion-based sampling methods. We consider four different methods: Time-Reversed Diffusion Sampler (DIS) (Berner et al., 2022), Monte Carlo Diffusions (MCD) (Doucet et al., 2022b), Controlled Monte Carlo Diffu-

| METHOD (X) | $f$ | $u^\theta$ | $v^\gamma$ |
|---|---|---|---|
| MCD | $\nabla \log \pi_t^\phi$ | ✗ | ✓ |
| CMCD[3] | $\nabla \log \pi_t^\phi$ | ✓ | ✓ |
| DIS | $\nabla \log p^\phi$ | ✓ | ✗ |
| DBS | ANY | ✓ | ✓ |

Table 1: Diffusion-based sampling methods considered in this work based on (2). Crosses indicate that the control is set to zero.

sions (CMCD) (Vargas et al., 2024) and Diffusion Bridge Sampler (DBS) (Richter et al., 2023). A summary is shown in Table 1. It is worth noting that we do not separately consider the Denoising Diffusion Sampler (DDS) (Vargas et al., 2023a), as it can be viewed as a special case of DIS. For reference, we consider Gaussian (GVI) and Gaussian mixture (GMVI) mean-field approximations (Wainwright & Jordan, 2008), both of which are special cases of the aforementioned methods for $N = 0$ diffusion steps with $K = 1$, $K \geq 1$, respectively (cf. Appendix B). Lastly, we consider three competing state-of-the-art methods, namely, Sequential Monte Carlo (SMC) (Del Moral et al., 2006), Continual Repeated Annealed Flow Transport (CRAFT) (Matthews et al., 2022), and Flow Annealed Importance Sampling Bootstrap (FAB) (Midgley et al., 2022).

For evaluation, we consider the effective sample size (ESS) and the marginal or extended evidence lower bound as performance criteria. Both are denoted as 'ELBO' for convenience. Next, if the ground truth normalization constant $\mathcal{Z}$ is available, we use an importance-weighted estimate $\hat{\mathcal{Z}}$ to

---

[3]Vargas et al. (2024) use the same in control in (2a) and (2b) by leveraging Nelson's relation (Nelson, 2020).

compute the estimation error $\Delta \log \mathcal{Z} = |\log \mathcal{Z} - \log \hat{\mathcal{Z}}|$. Additionally, if samples from the target $\pi$ are available, we compute the Sinkhorn distance $\mathcal{W}_\gamma^2$ (Cuturi, 2013).

To ensure a fair comparison, all experiments are conducted under identical settings. Our evaluation methodology adheres to the protocol by Blessing et al. (2024). For a comprehensive overview of the experimental setup see Appendix C. Moreover, a comprehensive set of ablation studies and additional experiments, are provided in Appendix D.

| | **Funnel** ($d = 10$) | | | |
|---|---|---|---|---|
| **Method** | ELBO ↑ | $\Delta \log \mathcal{Z}$ ↓ | ESS ↑ | $\mathcal{W}_\gamma^2$ ↓ |
| GVI | $-1.841_{\pm 0.003}$ | $0.691_{\pm 0.070}$ | $0.092_{\pm 0.006}$ | $178.007_{\pm 0.164}$ |
| GMVI | $\underline{-0.212_{\pm 0.001}}$ | $\underline{0.056_{\pm 0.004}}$ | $\underline{0.744_{\pm 0.018}}$ | $\underline{102.826_{\pm 0.109}}$ |
| MCD | $-0.721_{\pm 0.003}$ | $0.201_{\pm 0.017}$ | $0.207_{\pm 0.012}$ | $164.882_{\pm 0.363}$ |
| MCD-GP | $-0.724_{\pm 0.003}$ | $0.173_{\pm 0.046}$ | $0.206_{\pm 0.026}$ | $164.967_{\pm 0.334}$ |
| MCD-GMP | $\underline{-0.059_{\pm 0.002}}$ | $\underline{0.014_{\pm 0.001}}$ | $\underline{0.922_{\pm 0.012}}$ | $\underline{100.174_{\pm 0.174}}$ |
| CMCD | $-0.210_{\pm 0.002}$ | $0.020_{\pm 0.006}$ | $0.588_{\pm 0.013}$ | $104.652_{\pm 0.593}$ |
| CMCD-GP | $-0.211_{\pm 0.002}$ | $0.023_{\pm 0.003}$ | $0.567_{\pm 0.023}$ | $104.644_{\pm 0.710}$ |
| CMCD-GMP | $\underline{-0.027_{\pm 0.001}}$ | $\underline{0.005_{\pm 0.000}}$ | $\mathbf{0.950_{\pm 0.004}}$ | $102.027_{\pm 0.200}$ |
| DIS | $-0.286_{\pm 0.002}$ | $0.041_{\pm 0.008}$ | $0.483_{\pm 0.025}$ | $107.458_{\pm 0.670}$ |
| DIS-GP | $-0.296_{\pm 0.002}$ | $0.047_{\pm 0.003}$ | $0.498_{\pm 0.021}$ | $107.458_{\pm 0.826}$ |
| DIS-GMP | $\underline{-0.058_{\pm 0.002}}$ | $\underline{0.019_{\pm 0.002}}$ | $\underline{0.929_{\pm 0.017}}$ | $\mathbf{100.093_{\pm 0.028}}$ |
| DBS | $-0.180_{\pm 0.004}$ | $0.019_{\pm 0.005}$ | $0.600_{\pm 0.014}$ | $102.964_{\pm 0.442}$ |
| DBS-GP | $-0.187_{\pm 0.003}$ | $0.021_{\pm 0.003}$ | $0.603_{\pm 0.014}$ | $102.653_{\pm 0.586}$ |
| DBS-GMP | $\underline{-0.047_{\pm 0.002}}$ | $\underline{0.012_{\pm 0.002}}$ | $\underline{0.949_{\pm 0.008}}$ | $100.230_{\pm 0.088}$ |
| SMC | $-0.242_{\pm 0.047}$ | $0.187_{\pm 0.054}$ | - | $149.353_{\pm 2.973}$ |
| CRAFT | $-0.027_{\pm 0.060}$ | $0.091_{\pm 0.018}$ | - | $\underline{134.335_{\pm 0.663}}$ |
| FAB | $\mathbf{-0.014_{\pm 0.003}}$ | $\mathbf{0.001_{\pm 0.000}}$ | - | $153.894_{\pm 3.916}$ |

**GVI:** $K = 1,\ N = 0$     **GMVI:** $K = 10,\ N = 0$

**DIS-GP:** $K = 1,\ N = 128$     **DIS-GMP:** $K = 10,\ N = 128$

$K$: Num. mixture components     $N$: Num. diffusion steps

Figure 3: **Left side**: Results for Funnel target, averaged across four seeds. Evaluation criteria include evidence lower bound ELBO, importance-weighted errors for estimating the log-normalizing constant $\Delta \log \mathcal{Z}$, effective sample size ESS, Sinkhorn distance $\mathcal{W}_2^\gamma$. The best overall results are highlighted in bold, with category-specific best results underlined. Arrows (↑, ↓) indicate whether higher or lower values are preferable, respectively. Blue and green shading indicate that the method uses learned Gaussian (GP) and Gaussian mixture priors (GMP), respectively. Red shading indicate competing state-of-the-art methods. Note that ESS cannot be computed due to the use of resampling schemes. **Right side**: Visualization of the first two dimensions of the Funnel target. Colored ellipses and circles denote standard deviations and means of the Gaussian components, respectively. Red dots illustrate samples of the model.

## 6.1 BENCHMARK PROBLEMS

We evaluate the different methods on various real-world and synthetic target densities.

**Real-World Densities.** We consider six real-world target densities: Four Bayesian inference tasks, where inference is performed over the parameters of a logistic regression model, namely *Credit* ($d = 25$), *Cancer* ($d = 31$), *Ionosphere* ($d = 35$), and *Sonar* ($d = 61$). Moreover, *Seeds* ($d = 26$) and *Brownian* ($d = 32$), where the goal is to perform inference over the parameters of a random effect regression model, and the time discretization of a Brownian motion, respectively. For these densities, we do not have access to the ground truth normalizer $Z$ or samples from $\pi$ preventing us from computing errors for log normalization estimation $\Delta \log \mathcal{Z}$ and Sinkhorn distances $\mathcal{W}_\gamma^2$. The resulting ELBO values are presented in Table 2.

**Synthetic Densities.** The *Funnel* density was introduced by Neal (2003) as has a shape that resembles a funnel, where one part is tight and highly concentrated, while the other is spread out over a wide region, making it challenging for sampling algorithms to explore the distribution effectively. Next, we consider the *Fashion* target which uses NICE (Dinh et al., 2014) to train a normalizing flow on the high-dimensional $d = 28 \times 28 = 784$ MNIST Fashion dataset. A recent study by Blessing et al. (2024) showed that current state-of-the-art methods were not able to generate samples with high quality from multiple modes.

## 6.2 RESULTS

**Impact of Learned Gaussian (GP) and Gaussian Mixture (GMP) Priors.** We evaluated the performance of our proposed methods on both real-world tasks and the *Funnel* density, employing $N = 128$ diffusion steps across all methods and $K = 10$ mixture components for X-GMP. To ensure a fair comparison, we initialized the priors of all diffusion-based methods with zero mean

| METHOD | CREDIT | SEEDS | CANCER | BROWNIAN | IONOSPHERE | SONAR |
|---|---|---|---|---|---|---|
| GVI | $-605.561_{\pm0.166}$ | $-76.741_{\pm0.007}$ | $-147.453_{\pm0.144}$ | $-3.885_{\pm0.005}$ | $-123.391_{\pm0.013}$ | $-137.696_{\pm0.043}$ |
| GMVI | $-603.424_{\pm0.154}$ | $-75.221_{\pm0.011}$ | $-145.456_{\pm0.254}$ | $-2.250_{\pm0.011}$ | $-122.019_{\pm0.019}$ | $-135.959_{\pm0.031}$ |
| MCD | $-1399.241_{\pm497.114}$ | $-75.699_{\pm0.015}$ | $-148.471_{\pm8.565}$ | $-15.498_{\pm0.158}$ | $-114.320_{\pm0.007}$ | $-112.639_{\pm0.025}$ |
| MCD-GP | $-585.350_{\pm0.015}$ | $-73.542_{\pm0.003}$ | $-89.676_{\pm0.189}$ | $0.771_{\pm0.008}$ | $-111.897_{\pm0.004}$ | $-109.338_{\pm0.004}$ |
| MCD-GMP | $-585.276_{\pm0.013}$ | $-73.461_{\pm0.004}$ | $-88.562_{\pm0.243}$ | $0.993_{\pm0.003}$ | $-111.827_{\pm0.007}$ | $-109.197_{\pm0.004}$ |
| CMCD | $-586.956_{\pm0.018}$ | $-74.033_{\pm0.010}$ | $-80.076_{\pm0.118}$ | $-1.346_{\pm0.013}$ | $-112.183_{\pm0.006}$ | $-109.332_{\pm0.006}$ |
| CMCD-GP | $-585.178_{\pm0.013}$ | $-73.456_{\pm0.003}$ | $-78.576_{\pm0.068}$ | $1.043_{\pm0.005}$ | $-111.687_{\pm0.003}$ | $-108.669_{\pm0.007}$ |
| CMCD-GMP | $-585.162_{\pm0.002}$ | $-73.429_{\pm0.002}$ | $-78.402_{\pm0.037}$ | $1.087_{\pm0.001}$ | $-111.682_{\pm0.000}$ | $-108.634_{\pm0.000}$ |
| DIS | $-589.636_{\pm0.757}$ | $-74.400_{\pm0.007}$ | $-86.592_{\pm2.107}$ | $-3.503_{\pm0.019}$ | $-112.525_{\pm0.008}$ | $-110.153_{\pm0.022}$ |
| DIS-GP | $-585.247_{\pm0.009}$ | $-73.540_{\pm0.005}$ | $-85.005_{\pm1.286}$ | $0.588_{\pm0.013}$ | $-111.847_{\pm0.006}$ | $-109.280_{\pm0.024}$ |
| DIS-GMP | $-585.223_{\pm0.006}$ | $-73.492_{\pm0.003}$ | $-84.061_{\pm2.117}$ | $0.885_{\pm0.005}$ | $-111.811_{\pm0.002}$ | $-109.157_{\pm0.000}$ |
| DBS | $-587.366_{\pm0.683}$ | $-73.918_{\pm0.008}$ | $-82.466_{\pm4.090}$ | $-0.773_{\pm0.010}$ | $-112.070_{\pm0.005}$ | $-109.188_{\pm0.005}$ |
| DBS-GP | $-585.524_{\pm0.414}$ | $-73.437_{\pm0.001}$ | $-83.395_{\pm4.184}$ | $1.081_{\pm0.004}$ | $-111.673_{\pm0.002}$ | $-108.595_{\pm0.006}$ |
| DBS-GMP | $-585.148_{\pm0.002}$ | $\mathbf{-73.418_{\pm0.001}}$ | $-78.160_{\pm0.063}$ | $\mathbf{1.118_{\pm0.002}}$ | $\mathbf{-111.657_{\pm0.002}}$ | $\mathbf{-108.548_{\pm0.000}}$ |
| SMC | $-698.403_{\pm4.146}$ | $-74.699_{\pm0.100}$ | $-194.059_{\pm0.613}$ | $-1.874_{\pm0.622}$ | $-114.751_{\pm0.238}$ | $-111.355_{\pm1.177}$ |
| CRAFT | $-594.795_{\pm0.411}$ | $-73.793_{\pm0.015}$ | $-95.737_{\pm1.067}$ | $0.886_{\pm0.053}$ | $-112.386_{\pm0.182}$ | $-115.618_{\pm1.316}$ |
| FAB | $\mathbf{-585.102_{\pm0.001}}$ | $-73.418_{\pm0.002}$ | $-78.287_{\pm0.835}$ | $1.031_{\pm0.010}$ | $-111.678_{\pm0.003}$ | $-108.593_{\pm0.008}$ |

Table 2: Evidence lower bound (ELBO) values for various real-world benchmark problems, averaged across four seeds. The best overall results are highlighted in bold, with category-specific best results underlined. Blue and green shading indicate that the method uses learned Gaussian (GP) and Gaussian mixture priors (GMP), respectively. Red shading indicate competing state-of-the-art methods.

and unit variance. Table 2 and Figure 3 present our findings. The analysis demonstrates that GP consistently achieves tighter ELBO values compared to fixed priors, with GMP yielding further improvements over GP. Furthermore, Figure 3 illustrates both qualitatively and quantitatively that GMP effectively combines the strengths of Gaussian mixture and diffusion models, resulting in significant improvements. Specifically, we observed that the Gaussian components adapt well to the target's support, covering both the neck and opening of the funnel shape. This results in less complex dynamics and better target coverage for DIS-GMP compared to using a single Gaussian (DIS-GP). Notably, the combination of DBS and GMP outperforms state-of-the-art methods across the majority of tasks and evaluation metrics.

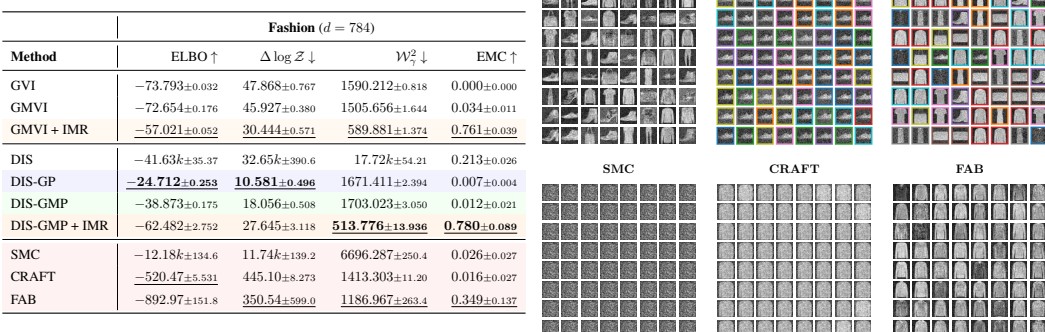

| | Fashion ($d = 784$) | | | |
|---|---|---|---|---|
| Method | ELBO $\uparrow$ | $\Delta \log \mathcal{Z} \downarrow$ | $\mathcal{W}_\gamma^2 \downarrow$ | EMC $\uparrow$ |
| GVI | $-73.793_{\pm0.032}$ | $47.868_{\pm0.767}$ | $1590.212_{\pm0.818}$ | $0.000_{\pm0.000}$ |
| GMVI | $-72.654_{\pm0.176}$ | $45.927_{\pm0.380}$ | $1505.656_{\pm1.644}$ | $0.034_{\pm0.011}$ |
| GMVI + IMR | $-57.021_{\pm0.052}$ | $30.444_{\pm0.571}$ | $589.881_{\pm1.374}$ | $0.761_{\pm0.039}$ |
| DIS | $-41.63k_{\pm35.37}$ | $32.65k_{\pm390.6}$ | $17.72k_{\pm54.21}$ | $0.213_{\pm0.026}$ |
| DIS-GP | $\mathbf{-24.712_{\pm0.253}}$ | $\mathbf{10.581_{\pm0.496}}$ | $1671.411_{\pm2.394}$ | $0.007_{\pm0.004}$ |
| DIS-GMP | $-38.873_{\pm0.175}$ | $18.056_{\pm0.508}$ | $1703.023_{\pm3.050}$ | $0.012_{\pm0.021}$ |
| DIS-GMP + IMR | $-62.482_{\pm2.752}$ | $27.645_{\pm3.118}$ | $\mathbf{513.776_{\pm13.936}}$ | $\mathbf{0.780_{\pm0.089}}$ |
| SMC | $-12.18k_{\pm134.6}$ | $11.74k_{\pm139.2}$ | $6696.287_{\pm250.4}$ | $0.026_{\pm0.027}$ |
| CRAFT | $-520.47_{\pm5.531}$ | $445.10_{\pm8.273}$ | $1413.303_{\pm11.20}$ | $0.016_{\pm0.027}$ |
| FAB | $-892.97_{\pm151.8}$ | $350.54_{\pm599.0}$ | $1186.967_{\pm263.4}$ | $0.349_{\pm0.137}$ |

Figure 4: **Left side**: Results for Fashion target, averaged across four seeds. Evaluation criteria include evidence lower bound ELBO, importance-weighted errors for estimating the log-normalizing constant $\Delta \log \mathcal{Z}$, and Sinkhorn distance $\mathcal{W}_2^\gamma$. The best overall results are highlighted in bold, with category-specific best results underlined. Arrows ($\uparrow, \downarrow$) indicate whether higher or lower values are preferable, respectively. Blue and green shading indicate that the method uses learned Gaussian (GP) and Gaussian mixture priors (GMP), respectively. Orange shading indicates that the method uses iterative model refinement (IMR). Red shading indicate competing state-of-the-art methods. **Right side**: Visualization of the $d = 28 \times 28 = 784$ dimensional Fashion samples. Top left corner visualizes samples from the target distribution. Colored frames indicate samples from different components of the Gaussian mixture.

**Ablation Study: Number of Mixture Components $K$ and Diffusion Steps $N$.** We further investigated the effect of varying the number of diffusion steps $N$ and mixture components $K$ on a subset of tasks for DIS. The results, shown in Figure 5, demonstrate consistent improvements in effective sample size (ESS) with increases in both $K$ and $N$. Additionally, we consistently observed that the combination of a higher number of components and diffusion steps yields the best overall performance. These trends hold across other metrics, as further detailed in Appendix D.

**Iterative Model Refinement (IMR).** Lastly, we investigated the impact of IMR, as detailed in Section 5, using DIS. For this analysis, we focused on the multi-modal Fashion target, which necessitates exploration in a high-dimensional space ($d = 784$). In addition to the performance criteria outlined in Section 6, we quantify how many of the modes the model discovered via the *entropic mode coverage (EMC)* introduced by Blessing et al. (2024). EMC evaluates the mode coverage of a sampler by leveraging prior knowledge of the target density. It holds that EMC $\in [0, 1]$ where EMC $= 1$ indicates that the model achieves uniform coverage over all modes whereas EMC $= 0$ indicates that the model only produces samples from a single mode. We employed the Metropolis-adjusted Langevin algorithm (MALA) (Cheng et al., 2018) to generate a set of candidate samples, noting that the computational cost of this process is comparable to a single gradient step in most diffusion-based sampling methods. The initial candidate samples as well as the support of DIS without learned prior are initialized such that they roughly cover the target support. Additional details are provided in Appendix C.2. We iteratively increased the number of components to $K = 10$, utilizing $N = 128$ diffusion steps throughout. Figure 4 presents our findings, demonstrating that the absence of IMR leads to mode collapse across all methods, as evidenced by high Sinkhorn distance values. The qualitative results highlight the role of candidate samples in facilitating mode discovery. Notably, the color-coding of DIS-GMP + IMR illustrates that each mixture component concentrates on a distinct mode, validating the effectiveness of the initialization heuristic proposed in (22) in balancing exploration and exploitation. This finding is also quantitatively reflected by the high EMC and low Sinkhorn distance values. In contrast, the ELBO and $\Delta \log \mathcal{Z}$ values are slightly worse when using GMPs and IMR. This is attributed to the fact that these performance criteria are not well-suited for quantifying the model performance for multi-modal targets and tend to favor models that fit a single mode perfectly (Blessing et al., 2024). Moreover, with higher $K$, the diffusion model has to learn more complex control functions, as it needs to operate over the support of the entire Gaussian Mixture Model (GMM) rather than a single Gaussian. This added complexity can introduce more opportunities for approximation errors, which may negatively impact ELBO and $\Delta \log \mathcal{Z}$ values compared to using a single learnable Gaussian. Nevertheless, the resulting samples from DIS-GMP are closer to the target distribution in terms of Sinkhorn distance. Importantly, these errors remain significantly smaller than those observed with non-learnable priors.

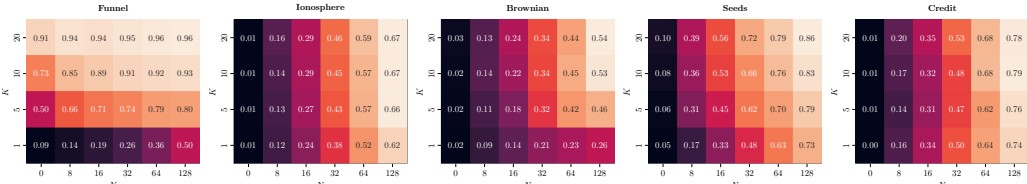

Figure 5: Effective sample size (ESS) of DIS-GMP for various real-world benchmark problems, averaged across four seeds. Here, $N$ denotes the number of discretization steps and $K$ the number of components in den Gaussian mixture.

## 7 CONCLUSION AND FUTURE WORK

In this paper, we propose a novel approach for improving diffusion-based sampling techniques by introducing end-to-end learnable Gaussian Mixture Priors (GMPs). Our method addresses key challenges in diffusion models—namely, large discretization errors, mode collapse, and poor exploration—by providing more expressive and adaptable priors compared to the conventional Gaussian priors. We conducted comprehensive experiments on both synthetic and real-world datasets, which consistently demonstrated the superior performance of our proposed method. The results underscore the effectiveness of GMPs in overcoming the limitations of traditional diffusion models while requiring little to no hyperparameter tuning. Furthermore, we developed a novel strategy for iterative model refinement, which involves progressively adding components to the mixture during training, and demonstrated its effectiveness on a challenging high-dimensional problem.

A promising direction for future research is the improvement of the iterative model refinement strategy. While we showed that progressively increasing the number of components in the Gaussian mixture improves performance, optimizing the selection criteria for adding new components, generating better candidate samples, or dynamically adjusting the number of components during training, could lead to further gains in efficiency and accuracy.

ACKNOWLEDGMENTS AND DISCLOSURE OF FUNDING

D.B. acknowledges support by funding from the pilot program Core Informatics of the Helmholtz Association (HGF) and the state of Baden-Württemberg through bwHPC, as well as the HoreKa supercomputer funded by the Ministry of Science, Research and the Arts Baden-Württemberg and by the German Federal Ministry of Education and Research.

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

## A  PROOFS

### A.1  PROOF OF PROPOSITION 1

We will use the Fokker-Planck equation (FPE) and show that the given stationary distribution satisfies it when the time derivative is set to zero:

First, recall the FPE for the probability density $p(x, t)$ of a process described by the stochastic differential equation (SDE)

$$\mathrm{d}x_t = -f(x_t)\mathrm{d}t + \sigma \mathrm{d}\mathbf{w}_t \tag{23}$$

is given as

$$\frac{\partial p(x, t)}{\partial t} = \nabla \cdot [f(x)p(x, t)] + \frac{\sigma^2}{2}\nabla^2 p(x, t). \tag{24}$$

For the stationary distribution $p^{\mathrm{st}}(x)$, we set $\frac{\partial p(x,t)}{\partial t} = 0$:

$$0 = \nabla \cdot [f(x)p^{\mathrm{st}}(x)] + \frac{\sigma^2}{2}\nabla^2 p^{\mathrm{st}}(x) \tag{25}$$

Next, recall the proposed stationary distribution:

$$p^{\mathrm{st}}(x) \propto \exp\left(-\frac{2}{\sigma^2}\int f(x)\mathrm{d}x\right) \tag{26}$$

Next, we verify that this satisfies stationary FPE ( (25)). First, let's compute the gradient and Laplacian of $p^{\mathrm{st}}(x)$:

$$\nabla p^{\mathrm{st}}(x) = p^{\mathrm{st}}(x) \cdot \left(-\frac{2}{\sigma^2}\right)f(x) \tag{27}$$

$$\nabla^2 p^{\mathrm{st}}(x) = \nabla \cdot [p^{\mathrm{st}}(x) \cdot \left(-\frac{2}{\sigma^2}\right)f(x)] \tag{28}$$

$$= p^{\mathrm{st}}(x) \cdot \left(-\frac{2}{\sigma^2}\right)^2 [f(x)]^2 + p^{\mathrm{st}}(x) \cdot \left(-\frac{2}{\sigma^2}\right)\nabla \cdot f(x) \tag{29}$$

---

**Algorithm 1** Training of diffusion sampler with learnable prior

**Require:**

- control functions $u_\theta, v_\gamma$ with initial parameters $\theta_0, \gamma_0$

- prior distribution $p_0^\phi$ with initial parameters $\phi_0$

- initial discretization step size $\delta t_0$

- number of gradient steps $G$, number of diffusion steps $N$, step size $\eta$

$\Theta_0 = \{\theta_0, \gamma_0, \phi_0, \delta t_0\}$
**for** $i \leftarrow 0, \dots, G-1$ **do**
    $x_0 \leftarrow g(\xi, \phi_i), \quad \xi \sim p(\cdot)$             $\triangleright$ sample $p_i^\phi$ via reparameterization (batched in practice)
    $\mathcal{L} \leftarrow \log p^{\phi_i}(x_0)$
    **for** $n \leftarrow 0, \dots, N-1$ **do**
        $x_{n+1} = x_n + \left[ f(x_n, n) + \sigma u^{\theta_i}(x_n, n) \right] \delta t_i + \sigma \sqrt{2\delta t_i} \epsilon_n$
        $\mathcal{L} \leftarrow \mathcal{L} + \log \vec{p}_{n+1|n}^{\theta_i, \phi_i}(x_{n+1}|x_n) - \log \overleftarrow{p}_{n|n+1}^{\gamma_i, \phi_i}(x_n|x_{n+1})$
    $\mathcal{L} \leftarrow \mathcal{L} - \log \rho(x_N)$
    $\Theta_{i+1} \leftarrow \Theta_i + \eta \nabla_\Theta \mathcal{L}$                    $\triangleright$ maximize (extended) ELBO
**return** optimized parameters $\Theta_G$

---

Finally, we substitute these into the left side of (25), that is,

$$\nabla \cdot [f(x)p^{\text{st}}(x)] + \frac{\sigma^2}{2} \nabla^2 p^{\text{st}}(x) \tag{30}$$

$$= \nabla \cdot [f(x)p^{\text{st}}(x)] + \frac{\sigma^2}{2} \left[ p^{\text{st}}(x) \cdot \left( -\frac{2}{\sigma^2} \right)^2 [f(x)]^2 + p^{\text{st}}(x) \cdot \left( -\frac{2}{\sigma^2} \right) \nabla \cdot f(x) \right] \tag{31}$$

$$= p^{\text{st}}(x)\nabla \cdot f(x) + f(x)\nabla p^{\text{st}}(x) + p^{\text{st}}(x) \left[ -\frac{2}{\sigma^2} \right] [f(x)]^2 - p^{\text{st}}(x)\nabla \cdot f(x) \tag{32}$$

$$= p^{\text{st}}(x)\nabla \cdot f(x) + f(x)p^{\text{st}}(x) \left( -\frac{2}{\sigma^2} \right) f(x) + p^{\text{st}}(x) \left[ -\frac{2}{\sigma^2} \right] [f(x)]^2 - p^{\text{st}}(x)\nabla \cdot f(x) \tag{33}$$

$$= 0, \tag{34}$$

which yields the desired result. $\qquad\qquad\qquad\qquad\qquad\qquad\qquad\qquad\qquad\qquad\qquad\qquad\qquad\square$

## B    ADDITIONAL DETAILS FOR DIFFUSION-BASED SAMPLER

**Pseudocode:** We additionally provide pseudocode in Algorithm 1 for a generic diffusion sampler with learnable prior $p_0^\phi$. For clarity, we present an update step for a single sample. In practice, however, one would use mini-batches for these updates.

**Special Cases of X-GMP:** Consider the generic (extended) ELBO for X-GMP, that is,

$$\mathcal{L}_{\text{GMP}}(\theta, \gamma, \phi) = \mathbb{E}_{x_{0:N} \sim \vec{p}^{\,\theta, \phi}} \left[ \log \frac{\rho(x_N)}{\sum_{k=1}^{K} \alpha_k p_0^{\phi_k}(x_0)} + \sum_{n=1}^{N} \log \frac{\overleftarrow{p}_{n-1|n}^{\gamma, \phi}(x_{n-1}|x_n)}{\vec{p}_{n|n-1}^{\theta, \phi}(x_n|x_{n-1})} \right], \tag{35}$$

with $p_0^{\phi_k}(x_0) = \mathcal{N}(x_0|\mu_k, \Sigma_k)$. We obtain the following special cases:

- **GVI** ($K = 1, N = 0$)**:** For a single Gaussian mixture component and zero diffusion steps, Equation (35) reduces to the marginal ELBO objective in Equation (9) for a Gaussian distribution, that is,

$$\mathcal{L}_{\text{GVI}}(\phi) = \mathbb{E}_{x_0 \sim p_0^\phi} \left[ \log \frac{\rho(x_0)}{p_0^\phi(x_0)} \right]. \tag{36}$$

- **GMMVI** ($K > 1, N = 0$)**:** Similarly, if we have zero diffusion steps, but multiple Gaussian mixture components we obtain the marginal ELBO for Gaussian mixture models, i.e.,

$$\mathcal{L}_{\text{GMVI}}(\phi) = \mathbb{E}_{x_0 \sim p_0^\phi} \left[ \log \frac{\rho(x_0)}{\sum_{k=1}^K \alpha_k p_0^{\phi_k}(x_0)} \right]. \tag{37}$$

  Please note that there are more sophisticated methods to train Gaussian mixture models for VI, see Arenz et al. (2018; 2022).

- **X-GP** ($K = 1, N > 0$)**:** For a single mixture component and multiple diffusion steps, we obtain the objective for X-GP, i.e., for a diffusion-model with learned Gaussian prior, given by

$$\mathcal{L}_{\text{GP}}(\theta, \gamma, \phi) = \mathbb{E}_{x_{0:N} \sim \vec{p}^{\,\theta,\phi}} \left[ \log \frac{\rho(x_N)}{p_0^\phi(x_0)} + \sum_{n=1}^N \log \frac{\overleftarrow{p}_{n-1|n}^{\gamma,\phi}(x_{n-1}|x_n)}{\vec{p}_{n|n-1}^{\theta,\phi}(x_n|x_{n-1})} \right]. \tag{38}$$

- **X-GMP** ($K > 1, N > 0$)**:** Having multiple multiple mixture components $K$ and diffusion steps $N$ results in the full X-GMP objective, as in Equation (35).

**Forward and Backward Transitions.** We provide further information about the diffusion-based sampling methods considered in this work in Table 3. Specifically, we provide expressions for the forward and backward transitions.

**Time complexity.** Diffusion-based samplers that use a Gaussian prior have a time complexity of $\mathcal{O}(N)$, whereas Gaussian Mixture Priors (GMPs) incur a time complexity of $\mathcal{O}(NK)$. The additional factor $K$ arises from the need to compute the likelihood of the GMP at each diffusion step. However, in practice, the evaluation of the likelihood of the GMP can be parallelized across its components, which substantially reduces the computational overhead. This parallelization allows for efficient implementation despite the increased theoretical complexity.

**Memory consumption.** When using the standard "discrete-then-optimize" approach to minimize the KL divergence in (11), which requires differentiation through the SDE, memory consumption scales linearly with both $K$ (number of components) and $N$ (number of diffusion steps). In contrast, methods like the stochastic adjoint approach for KL optimization (Li et al., 2020) achieve constant memory consumption, making them more suitable for scenarios with a large number of components or diffusion steps.

In our experiments, we opted for the former approach due to its simplicity. However, for tasks involving extensive components or steps, the stochastic adjoint method or similar approaches may be more practical. Additionally, constant memory consumption can also be achieved by using alternative loss functions such as the log-variance loss (Richter et al., 2020; 2023) or moment-loss (Hartmann et al., 2019).

**Initialization heuristic for iterative model refinement.** As an alternative to (22) one could consider the following heuristic for selecting the mean of a new component:

$$\mu_{\text{new}} = \arg\max_{x_0 \in \mathcal{C}} \mathbb{E}_{x_{1:N} \sim \vec{p}^{\,\theta,\phi}} \left[ \log \frac{\rho(x_N)}{p_0^\phi(x_0)} + \sum_{n=1}^N \log \frac{\overleftarrow{p}_{n-1|n}^{\gamma,\phi}(x_{n-1}|x_n)}{\vec{p}_{n|n-1}^{\theta,\phi}(x_n|x_{n-1})} \right]. \tag{39}$$

However, empirically we found that choosing (22) results in increased sample diversity. We leave an in-depth exploration of different heuristics as future work.

## C  EXPERIMENTAL DETAILS

### C.1  BENCHMARKING TARGETS

This section introduces the target densities considered in our experiments. Please note that the majority of tasks are taken from the recent benchmark study from Blessing et al. (2024). For convenience, we provide a brief explanation of the target densities.

| Method | $\vec{p}^{\,\theta,\phi}_{n+1|n}(x_{n+1}|x_n)$ | $\overleftarrow{p}^{\,\gamma,\phi}_{n-1|n}(x_{n-1}|x_n)$ |
|--------|------|------|
| DIS | $\mathcal{N}\left(x_{n+1}|x_n + \left[-\sigma^2\nabla\log p_0^\phi(x_n) + \sigma u^\theta(x_n,n)\right]\delta t, 2\sigma^2\delta tI\right)$ | $\mathcal{N}\left(x_{n-1}|x_n + \sigma^2\nabla\log p_0^\phi(x_n)\delta t, 2\sigma^2\delta tI\right)$ |
| MCD | $\mathcal{N}\left(x_{n+1}|x_n + \sigma^2\nabla_x\log\pi_n^\phi(x_n)\delta t, 2\sigma^2\delta tI\right)$ | $\mathcal{N}\left(x_{n-1}|x_n - \left[\sigma^2\nabla_x\log\pi_n^\phi(x_n) - \sigma v^\gamma(x_n,n)\right]\delta t, 2\sigma^2\delta tI\right)$ |
| CMCD | $\mathcal{N}\left(x_{n+1}|x_n + \left[\sigma^2\nabla_x\log\pi_n^\phi(x_n) + \sigma u^\theta(x_n,n)\right]\delta t, 2\sigma^2\delta tI\right)$ | $\mathcal{N}\left(x_{n-1}|x_n + \left[\sigma^2\nabla_x\log\pi_n^\phi(x_n) - \sigma u^\theta(x_n,n)\right]\delta t, 2\sigma^2\delta tI\right)$ |
| DBS | $\mathcal{N}\left(x_{n+1}|x_n + \left[f(x_n,n) + \sigma u^\theta(x_n,n)\right]\delta t, 2\sigma^2\delta tI\right)$ | $\mathcal{N}\left(x_{n-1}|x_n - \left[f(x_n,n) - \sigma v^\gamma(x_n,n)\right]\delta t, 2\sigma^2\delta tI\right)$ |

Table 3: Comparison of different forward and backward transitions $\vec{p}^{\,\theta,\phi}$, and $\overleftarrow{p}^{\,\gamma,\phi}$, respectively, for diffusion-based sampling methods based on $f$, $\pi_n^\phi$, $p_0^\phi$, $u^\theta$ and $v^\gamma$ as defined in the text.

**Bayesian Logistic Regression**: We evaluate a Bayesian logistic regression model on four standardized binary classification datasets:

- **Ionosphere** ($d = 35$, 351 $(x_i, y_i)$ pairs)
- **Sonar** ($d = 61$, 208 $(x_i, y_i)$ pairs)
- **German Credit** ($d = 25$, 1000 $(x_i, y_i)$ pairs)
- **Breast Cancer** ($d = 31$, 569 $(x_i, y_i)$ pairs)

The model assumes:

$$\omega \sim \mathcal{N}(0, \sigma_\omega^2 I),$$
$$y_i \sim \text{Bernoulli}(\text{sigmoid}(\omega^\top x_i)),$$

where features are standardized for linear logistic regression. Here, we perform inference over the parameters $\omega$ of the (linear) logistic regression model. In Blessing et al. (2024), the authors used an uninformative prior for the parameters of the Bayesian logistic regression models for the *Credit* and *Cancer* tasks, which frequently caused numerical instabilities. To maintain the challenge of the tasks while ensuring stability, we opted for a Gaussian prior with zero mean and variance of $\sigma_\omega^2 = 100$.

**Random Effect Regression**: We apply random effect regression to the **Seeds** dataset ($d = 26$):

$$\tau \sim \text{Gamma}(0.01, 0.01),$$
$$a_0, a_1, a_2, a_{12} \sim \mathcal{N}(0, 10),$$
$$b_i \sim \mathcal{N}(0, \frac{1}{\sqrt{\tau}}), \quad i = 1, \ldots, 21,$$
$$\text{logits}_i = a_0 + a_1 x_i + a_2 y_i + a_{12} x_i y_i + b_1,$$
$$r_i \sim \text{Binomial}(\text{logits}_i, N_i),$$

with inference conducted over model parameters given observed data.

**Time Series Models**: For time series analysis, we use the **Brownian** model ($d = 32$):

$$\alpha_{\text{inn}} \sim \text{LogNormal}(0, 2),$$
$$\alpha_{\text{obs}} \sim \text{LogNormal}(0, 2),$$
$$x_1 \sim \mathcal{N}(0, \alpha_{\text{inn}}),$$
$$x_i \sim \mathcal{N}(x_{i-1}, \alpha_{\text{inn}}), \quad i = 2, \ldots, 20,$$
$$y_i \sim \mathcal{N}(x_i, \alpha_{\text{obs}}), \quad i = 1, \ldots, 30,$$

with inference focusing on parameters $\alpha_{\text{inn}}, \alpha_{\text{obs}}$, and latent states $\{x_i\}_{i=1}^{30}$.

**Funnel**: ($d = 10$), a funnel-shaped distribution defined by:

$$\pi(x) = \mathcal{N}(x_1; 0, \sigma_f^2)\mathcal{N}(x_{2:10}; 0, \exp(x_1)I),$$

with $\sigma_f^2 = 9$.

**Fashion and Digits.** MNIST variants (**DIGITS**) and Fashion MNIST (**Fashion**) datasets using NICE (Dinh et al., 2014) to train normalizing flows, with resolutions $14 \times 14$ and DIGITS and $28 \times 28$ for Fashion.

## C.2 DIFFUSION-BASED METHODS: DETAILS AND TUNING

**General setting:** All experiments are conducted using the Jax library (Bradbury et al., 2021). Our default experimental setup, unless specified otherwise, is as follows: We use a batch size of 2000 (halved if memory-constrained) and train for $140k$ gradient steps to ensure approximate convergence. We use the Adam optimizer (Kingma & Ba, 2014), gradient clipping with a value of 1, and a learning rate scheduler that starts at $8 \times 10^{-3}$ and uses a cosine decay starting at 60k gradient steps. We utilized 128 discretization steps and the Euler-Maruyama method for integration. The control functions $u^\theta$ and $v^\gamma$ were parameterized as two-layer neural networks with 128 neurons. For DBS, we set the drift to $f = \sigma^2 \nabla \log \pi$.

Unlike Zhang & Chen (2021), we did not include the gradient of the target density in the network architecture. Inspired by Nichol & Dhariwal (2021), we applied a cosine-square scheduler for the discretization step size: $\delta t = a \cos^2\left(\frac{\pi}{2}\frac{n}{N}\right)$, where $a : [0, \infty) \to (0, \infty)$ is learned for all methods. We enforced non-negativity of $a$ via an element-wise softplus transformation. The diffusion coefficient $\sigma$ was set to 1 for all experiments. Furthermore, we set the initial $a$ to 0.1 for all experiments except Brownian, where we set 0.01. We did not perform any hyperparameter tuning since most parameters are learned end-to-end.

**Gaussian Priors (GP) and Gaussian Mixture Priors (GMP):** We learn diagonal Gaussian priors and ensure positive definiteness with an element-wise softplus transformation. We use a separate learning rate of $10^{-2}$ for all experiments to allow for quick adaptation of the Gaussian components. Furthermore, the mean was initialized at 0 and the initial covariance matrix was set to the identity except for Fashion where we set the initial variance to 5 which roughly covers the support of the target. The individual components in the Gaussian mixture follow the setup of Gaussian priors. The mixture weights are uniformly initialized and fixed during training. If not otherwise specified, we use $K = 10$ mixture components for X-GMP.

**Iterative Model Refinement (IMR):** For IMR, we add a new component after 500 training iterations starting with a single component. The initial means were selected with the heuristic presented in Equation (22). The variance of the newly added components was set to be 1. The candidate sample set was generated using the Metropolis Adjusted Langevin Algorithm (MALA) (Cheng et al., 2018). For that, we used 2000 random samples from a Gaussian with zero mean and variance 5, which roughly covers the support of the *Fashion* target. Please note that competing methods also use this prior knowledge for initialization of the prior, see Table 4. We use 128 steps steps, that is,

$$x_{i+1} = x_i + \tilde{\sigma}^2 \nabla \log \pi(x_i)\delta t + \tilde{\sigma}\sqrt{2\delta t}\epsilon, \quad \epsilon \sim \mathcal{N}(\cdot|0, I) \tag{40}$$

with $\tilde{\sigma} = 5$ and an additional Metropolis adjustment step. Here, $\tilde{\sigma}$ was chosen such that the final set of samples yields high target log-likelihoods $\log \rho(x)$. The final samples are used as candidate set. We note that this procedure brings the new components close to different modes in the target distribution and therefore facilitates exploration. Moreover, the computation of such a candidate set is very cheap, i.e., the equivalent of a single gradient step for e.g. MCD or CMCD.

## C.3 COMPETING METHODS: DETAILS AND TUNING

The results for competing methods presented in this work are primarily drawn from Blessing et al. (2024), where hyperparameters were carefully optimized. For convenience, we repeat the details. Since our experimental setup differs for the *Credit* and *Cancer* tasks (detailed in Section C.1), we adhered to the tuning recommendations provided by Blessing et al. (2024). Details about hyperparameters can be found in Table 4.

**Sequential Monte Carlo (SMC) and Continual Repeated Annealed Flow Transport (CRAFT):** The Sequential Monte Carlo (SMC) approach was implemented with 2000 particles and 128 annealing steps, matching the number of sequential steps used in diffusion-based sampling methods. Resampling was performed with a threshold of 0.3, and one Hamiltonian Monte Carlo (HMC) step was applied per temperature, using 5 leapfrog steps. The HMC step size was tuned according to Table 4, with different step sizes based on the annealing parameter $\beta_t$. Additionally, the scale of the initial proposal distribution was tuned. As CRAFT builds on the SMC framework, it used the same SMC specifications, incorporating diagonal affine flows (Papamakarios et al., 2021) as transition models.

| Methods / Parameters | Grid | Funnel | Fashion | Credit | Cancer | Brownian | Sonar | Seeds | Ionosphere | $\phi^4$ |
|---|---|---|---|---|---|---|---|---|---|---|
| **SMC** | | | | | | | | | | |
| Initial Scale | $\{0.1, 1, 10\}$ | 1 | $5^\dagger$ | 0.1 | 1 | 1 | 1 | 1 | 1 | $5^\dagger$ |
| HMC stepsize ($\beta \leq 0.5$) | $\{0.005, 0.001, 0.01, 0.05, 0.1, 0.2\}$ | 0.001 | 0.2 | 0.01 | 0.01 | 0.001 | 0.05 | 0.2 | 0.2 | 0.1 |
| HMC stepsize ($\beta > 0.5$) | $\{0.005, 0.001, 0.01, 0.05, 0.1, 0.2\}$ | 0.1 | 0.2 | 0.005 | 0.005 | 0.05 | 0.001 | 0.05 | 0.2 | 0.05 |
| **CRAFT** | | | | | | | | | | |
| Initial Scale | $\{0.1, 1, 10\}$ | 1 | $5^\dagger$ | 1 | 1 | 1 | 1 | 0.1 | 0.1 | |
| Learning Rate | $\{10^{-3}, 10^{-4}, 5 \times 10^{-4}, 10^{-5}\}$ | $10^{-3}$ | $10^{-4}$ | $5 \times 10^{-4}$ | $5 \times 10^{-4}$ | $10^{-3}$ | $10^{-3}$ | $10^{-3}$ | $10^{-3}$ | |
| **FAB** | | | | | | | | | | |
| Initial Scale | $\{0.1, 1, 10\}$ | 1 | $5^\dagger$ | 1 | 1 | 1 | 0.1 | 0.1 | 1 | |
| Learning Rate | $\{10^{-3}, 10^{-4}, 5 \times 10^{-4}, 10^{-5}\}$ | $10^{-4}$ | $10^{-3}$ | $10^{-4}$ | $10^{-3}$ | $10^{-3}$ | $10^{-3}$ | $10^{-4}$ | $10^{-3}$ | |

Table 4: Hyperparameter selection for all different sampling algorithms. The 'Grid' column indicates the values over which we performed a grid search. The values in the column which are marked with experiment names indicate which values were chosen for the reported results. The values for parameters indicated with † are set by using prior knowledge about the task.

**Flow Annealed Importance Sampling Bootstrap (FAB):** Automatic step size tuning for the SMC sampler was applied on top of the normalizing flow (Papamakarios et al., 2021). The flow architecture utilized RealNVP (Dinh et al., 2016), with an 8-layer MLP serving as the conditioner. FAB's replay buffer was employed to accelerate computations. The learning rate and base distribution scale were adjusted for target specificity as outlined in Table 4. A batch size of 2000 was used, and FAB was trained until reaching approximate convergence, which was sufficient to achieve approximate convergence.

## C.4 EVALUATION

**Evaluation protocol and model selection** We follow the evaluation protocol of prior work (Blessing et al., 2024) and evaluate all performance criteria 100 times during training, using 2000 samples for each evaluation. To smooth out short-term fluctuations and obtain more robust results within a single run, we apply a running average with a window of 5 evaluations. We conduct each experiment using four different random seeds and average the best results of each run.

**Performance Criteria:** In order to define the performance criteria, we first define the unnormalized (extended) importance weights $\tilde{w}$, that is,

$$\tilde{w} := \frac{\rho(x_N) \prod_{n=1}^N \overleftarrow{p}_{n-1|n}^{\gamma,\phi}(x_{n-1}|x_n)}{p_0^\phi(x_0) \prod_{n=1}^N \overrightarrow{p}_{n|n-1}^{\theta,\phi}(x_n|x_{n-1})}. \tag{41}$$

We consider the following following performance criteria:

- **Evidence lower bound (ELBO)**: We compute the (extended) ELBO as

$$\text{ELBO} := \mathbb{E}_{x_{0:N} \sim \overrightarrow{p}^{\,\theta,\phi}} [\log \tilde{w}] \approx \frac{1}{m} \sum_{i=1}^m \log \tilde{w}^{(i)}. \tag{42}$$

- **Estimation error** $\Delta \log \mathcal{Z}$: When having access to the ground truth normalization constant $\log \mathcal{Z}$, we can compute the estimation error $\Delta \log \mathcal{Z} = |\log \mathcal{Z} - \log \widehat{\mathcal{Z}}|$ using an importance weighted estimate, that is,

$$\log \widehat{\mathcal{Z}} := \log \mathbb{E}_{x_{0:N} \sim \overrightarrow{p}^{\,\theta,\phi}} [\tilde{w}] \approx \log \frac{1}{m} \sum_{i=1}^m \tilde{w}^{(i)}. \tag{43}$$

- **Effective sample size (ESS):** Moreover, we compute the (normalized) ESS as

$$\text{ESS} := \frac{\left(\sum_{i=1}^m \tilde{w}^{(i)}\right)^2}{m \sum_{i=1}^m \left(\tilde{w}^{(i)}\right)^2}. \tag{44}$$

- **Sinkhorn distance:** We estimate the Sinkhorn distance $\mathcal{W}_\gamma^2$ (Cuturi, 2013), i.e., an entropy regularized optimal transport distance between a set of samples from the model and target using the Jax `ott` library (Cuturi et al., 2022). Note that computing $\mathcal{W}_\gamma^2$ requires samples from the target density which are typically not available for real-world target densities.

| $K$ | DIS-GMP | | CMCD-GMP | |
|---|---|---|---|---|
| | ABS. [$s$] | REL. [%] | ABS. [$s$] | REL. [%] |
| 1 | 0.103 | - | 1.123 | - |
| 5 | 0.128 | 24.27 | 1.166 | 3.82 |
| 10 | 0.155 | 50.48 | 1.203 | 7.12 |

Table 5: Wallclock time of DIS-GMP and CMCD-GMP for the Fashion target for $N = 128$. Here, $K$ the number of components in den Gaussian mixture, 'abs.' denotes the absolute time per gradient step in seconds, and 'rel.' denotes the relative increase in percent compared to $K = 1$.

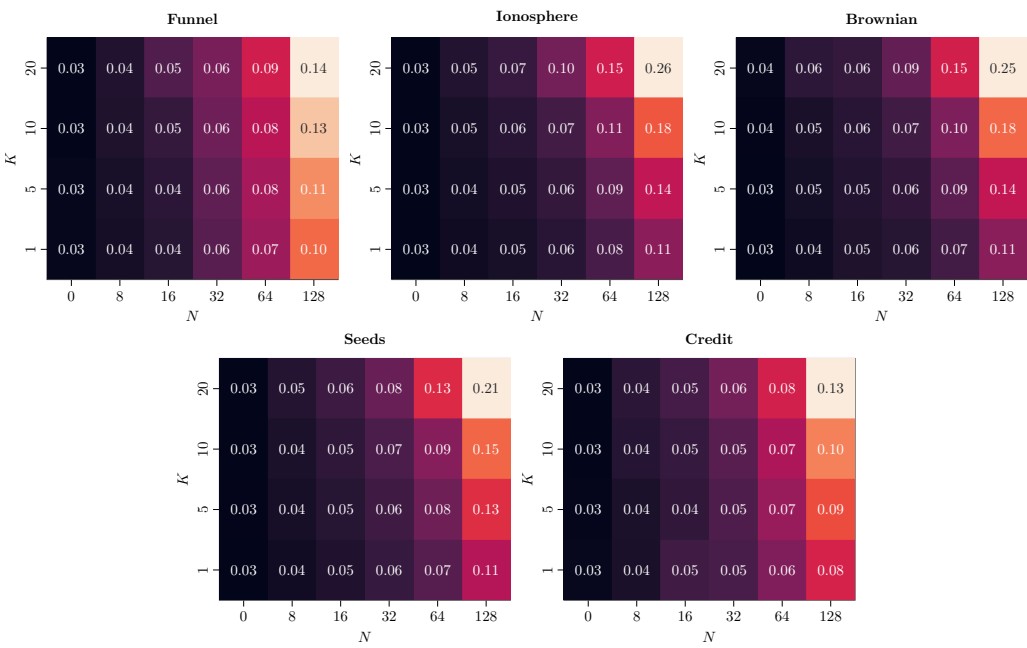

Figure 6: Wallclock time per gradient step of DIS-GMP for various benchmark problems. Here, $N$ denotes the number of discretization steps and $K$ the number of components in den Gaussian mixture.

- **Entropic mode coverage (EMC):** EMC evaluates the mode coverage of a sampler by leveraging prior knowledge of the target density. It holds that EMC $\in [0, 1]$ where EMC = 1 indicates that the model achieves uniform coverage over all modes whereas EMC = 0 indicates that the model only produces samples from a single mode. Please note that EMC does not provide any information about the sample quality. For further details, we refer the interested reader to Blessing et al. (2024).

## D   FURTHER NUMERICAL RESULTS

Here, we provide further numerical results.

**Wallclock time**   We further report the wallclock time per gradient step for DIS for a different number of diffusion steps $N$ and mixture components $K$. The results are shown in Figure 6. For $N \leq 64$, the Gaussian mixture prior barely influences the wallclock time where using $K = 10$ components roughly adds a 20 percent increase. Considering the performance improvements this is a good trade-off. For $N = 128$, Using $K = 10$ roughly results in a 50 percent increase as the likelihood of the prior has to be evaluated in every diffusion step. However, since most diffusion-based methods apart from DIS additionally require evaluating the target density at every step, the relative costs of using GMPs reduce if the target is more expensive to evaluate. We empirically validated this by additionally including a comparison between the wallclock time for DIS and CMCD on the Fashion target in Table 5. In this setting, the relative cost added by the GMP is minor.

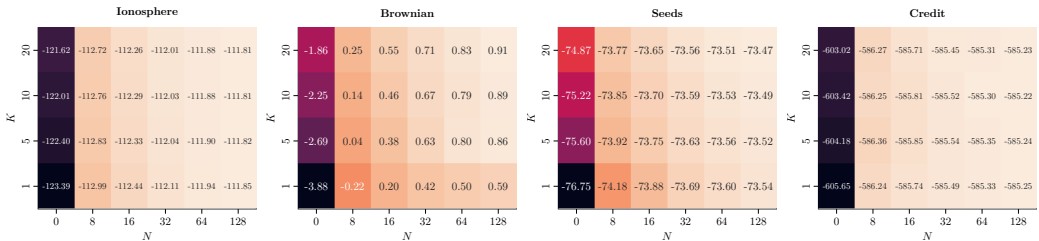

Figure 7: Evidence Lower Bound (ELBO) of DIS-GMP for various real-world benchmark problems, averaged across four seeds. Here, $N$ denotes the number of discretization steps and $K$ the number of components in den Gaussian mixture.

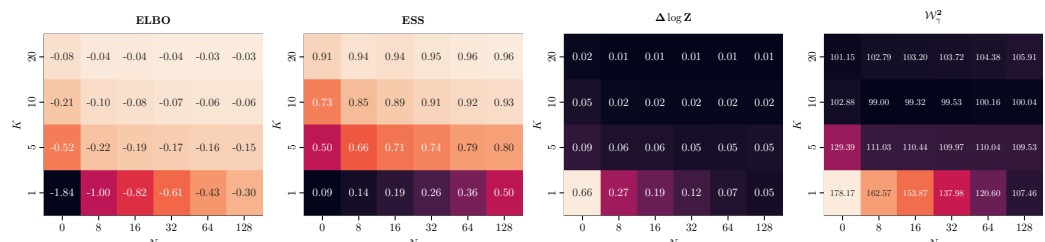

Figure 8: Various performance criteria of DIS-GMP for the Funnel target, averaged across four seeds. Here, $N$ denotes the number of discretization steps and $K$ the number of components in den Gaussian mixture.

**Additional results for DIS-GMP**    We present further details regarding the ablation study from Section 6. Specifically, we report ELBO values for the real-world benchmark problems in Figure 7 and various metrics for the *Funnel* target in Figure 8. The results are consistent with the results in Figure 5, where the performance improves with a higher number of mixture components $K$ and diffusion steps $N$.

**DIS-GMP on *Digits* target**    We additionally investigate the performance of DIS-GMP on the synthetic digits target. The results are reported in Figure 9. Here, we observe that the ELBO get looser when using more mixture components $K$, while $\Delta \log \mathcal{Z}$ stays roughly constant. However, the sample diversity improves significantly as shown quantitatively from the Sinkhorn distance $\mathcal{W}_\gamma^2$ and qualitatively in the Figure on the right-hand side.

|  | **Digits** ($d = 196$) | | |
|---|---|---|---|
| $K$ | ELBO $\uparrow$ | $\Delta \log \mathcal{Z} \downarrow$ | $\mathcal{W}_\gamma^2 \downarrow$ |
| 1 | $-\mathbf{12.090}_{\pm \mathbf{0.050}}$ | $5.269_{\pm 0.416}$ | $197.566_{\pm 0.340}$ |
| 5 | $-12.303_{\pm 0.350}$ | $\mathbf{4.419}_{\pm \mathbf{0.316}}$ | $183.241_{\pm 8.776}$ |
| 10 | $-13.820_{\pm 0.831}$ | $4.658_{\pm 0.260}$ | $164.827_{\pm 2.626}$ |
| 20 | $-15.413_{\pm 0.317}$ | $5.663_{\pm 0.085}$ | $\mathbf{151.006}_{\pm \mathbf{0.640}}$ |

$K = 1$    $K = 5$

$K = 10$    $K = 20$

$K$: Num. mixture components

Figure 9: **Left side**: Results for Digits target, averaged across four seeds using DIS-GMP+IMR. Evaluation criteria include evidence lower bound ELBO, importance-weighted errors for estimating the log-normalizing constant $\Delta \log \mathcal{Z}$, and Sinkhorn distance $\mathcal{W}_2^\gamma$. The best results are highlighted in bold. Arrows ($\uparrow$, $\downarrow$) indicate whether higher or lower values are preferable, respectively. **Right side**: Visualization of the $d = 14 \times 14 = 196$ dimensional Digits samples for a different number of mixture components $K$.

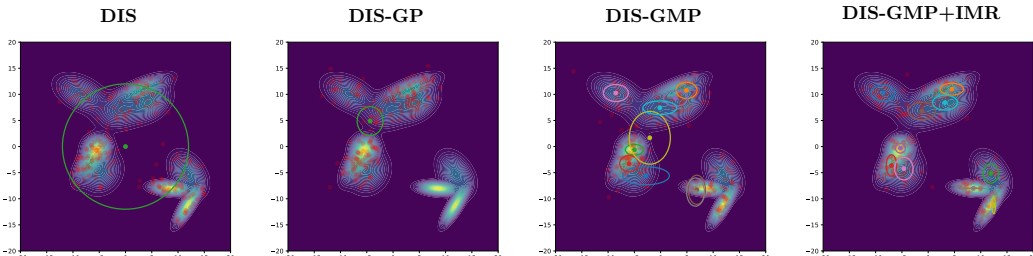

Figure 10: Visualization of a two-dimensional Gaussian mixture target density for different variants of DIS with $N = 128$ diffusion steps and $K = 10$ components for GMP versions. Colored ellipses and circles denote standard deviations and means of the Gaussian components, respectively. Red dots illustrate samples of the learned model.

| METHOD | $\nabla \log \pi$ | CREDIT | SEEDS | CANCER | BROWNIAN | IONOSPHERE | SONAR |
|---|---|---|---|---|---|---|---|
| DIS-GP | ✗ | $-585.247_{\pm 0.009}$ | $-73.540_{\pm 0.005}$ | $-85.005_{\pm 1.286}$ | $0.588_{\pm 0.013}$ | $-111.847_{\pm 0.006}$ | $-109.280_{\pm 0.024}$ |
| DIS-GP | ✓ | $-592.262_{\pm 0.794}$ | $\underline{-73.497_{\pm 0.001}}$ | $-96.180_{\pm 10.044}$ | N/A | $-111.957_{\pm 0.090}$ | $-109.473_{\pm 0.143}$ |
| DIS-GMP | ✗ | $\mathbf{-585.223_{\pm 0.006}}$ | $-73.492_{\pm 0.003}$ | $\mathbf{-84.061_{\pm 2.117}}$ | $\mathbf{0.885_{\pm 0.005}}$ | $\mathbf{-111.811_{\pm 0.002}}$ | $\mathbf{-109.157_{\pm 0.000}}$ |
| DIS-GMP | ✓ | $-586.817_{\pm 0.906}$ | $\mathbf{-73.475_{\pm 0.002}}$ | $-84.732_{\pm 0.466}$ | N/A | $-112.108_{\pm 0.002}$ | $-109.248_{\pm 0.001}$ |

Table 6: Evidence lower bound (ELBO) values for various real-world benchmark problems, averaged across four seeds. Here, $\nabla \log \pi$ indicates if the model architecture uses target score as described in Equation (46). The best overall results are highlighted in bold, with category-specific best results underlined. Blue and green shading indicate that the method uses learned Gaussian (GP) and Gaussian mixture priors (GMP), respectively.

**Ablation on Gaussian Mixture Target** We additionally experiment with using a two-dimensional Gaussian mixture model (GMM) as the target density. The GMM has ten components where the means are uniformly sampled in $[-12, 12]$ and the covariance matrices are sampled from a Wishart distribution. In addition to the performance criteria outlined in Section 6, we quantify the variation of the dynamics over time using the spectral norm of the Jacobian of the learned control, i.e.,

$$S = \mathbb{E}_{x_{0:T} \sim \vec{p}^{\,\theta}} \left[ \int_0^T \left\| \frac{\partial \sigma u^\theta(x, t)}{\partial x} \right\|_2 \mathrm{d}t \right]. \tag{45}$$

For DIS we initialized the prior with a standard deviation of 12 such that the prior covers the support of the target. For DIS-GMP, we use $K = 10$ components that are initialized with a standard deviation of 1. We report qualitative and qualitative results in Figure 10 and Figure 11. We find that DIS without learned prior and sufficiently large prior support is able to cover all modes as indicated by EMC $\approx 1$. While the sample quality is similar between DIS and GMP counterparts (see $\mathcal{W}_\gamma^2$), plain DIS needs significantly more diffusion steps in order to achieve similar ELBO/ESS values compared to the GMP counterparts which achieve good performance with as few as 8 diffusion steps. The requirement of plain DIS for more discretization steps is additionally reflected in the variation of the dynamics over time $S$. Lastly, the GP version is not able to cover all modes due to the mode-seeking nature of the reverse KL as indicated by the EMC and $\mathcal{W}_\gamma^2$ values.

**Ablation: Influence of the control architecture** We further evaluate the performance using the architecture by Zhang & Chen (2021) which additionally incorporates the score of the target, i.e. $\nabla \log \pi$, into the architecture via

$$u^\theta(x, t) = f_1^\theta(x, t) + f_2^\theta(t) \nabla \log \pi(\text{stop\_gradient}(x)). \tag{46}$$

where $f_1$ and $f_2$ are parameterized function approximatior with parameters $\theta$. Zhang & Chen (2021); Vargas et al. (2023a) found that detaching, that is, using a stop-gradient operator on $x$ yields superior results due to the simplification of the computational graph. We adopt this change and report the results in Table 6. We find that using the score of the target leads, in the majority of experiments, slightly worse results.

**Ablation: Kullback-Leibler vs. Log-Variance divergence** We further compare the KL divergence to the log-variance divergence introduced in Richter et al. (2020) and later extended to diffu-

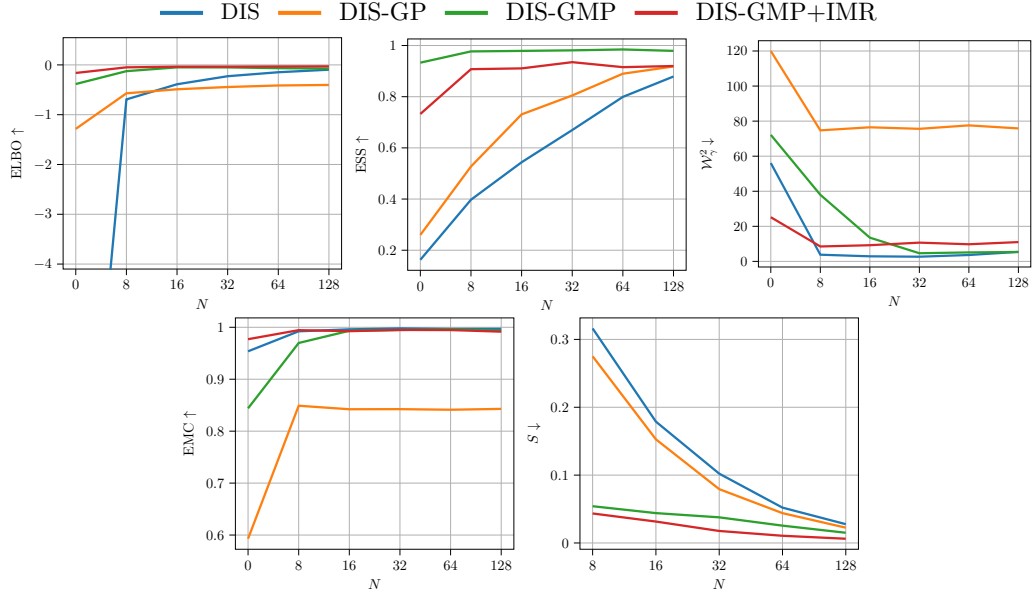

Figure 11: Results for the two-dimensional Gaussian mixture target, averaged across four seeds and reported across different numbers of diffusion steps $N$ for different variants of DIS. Evaluation criteria include evidence lower bound ELBO, importance-weighted errors for estimating the log-normalizing constant $\Delta \log \mathcal{Z}$, and Sinkhorn distance $\mathcal{W}_2^\gamma$, entropic mode coverage EMC, and the time-integrated spectral norm of the control $S$ (see Equation (45)).

| METHOD | DIV. | CREDIT | SEEDS | CANCER | BROWNIAN | IONOSPHERE | SONAR |
|---|---|---|---|---|---|---|---|
| DIS | KL | $-589.636_{\pm 0.757}$ | $-74.400_{\pm 0.007}$ | $-86.592_{\pm 2.107}$ | $-3.503_{\pm 0.019}$ | $-112.525_{\pm 0.008}$ | $-110.153_{\pm 0.022}$ |
| DIS | LV | $-5170.845_{\pm 5.627}$ | $-74.654_{\pm 0.022}$ | $-88.379_{\pm 1.491}$ | $-5.682_{\pm 0.303}$ | $-112.609_{\pm 0.053}$ | $-110.622_{\pm 0.071}$ |
| DIS-GP | KL | $-585.247_{\pm 0.009}$ | $-73.540_{\pm 0.005}$ | $-85.005_{\pm 1.286}$ | $0.588_{\pm 0.013}$ | $-111.847_{\pm 0.006}$ | $-109.280_{\pm 0.024}$ |
| DIS-GP | LV | $-5163.451_{\pm 3.296}$ | $-73.703_{\pm 0.177}$ | $-549.071_{\pm 466.902}$ | $\underline{0.729_{\pm 0.004}}$ | $\underline{-111.839_{\pm 0.006}}$ | $-109.498_{\pm 0.005}$ |
| DIS-GMP | KL | $-585.223_{\pm 0.006}$ | $\underline{-73.492_{\pm 0.003}}$ | $\underline{-84.061_{\pm 2.117}}$ | $0.885_{\pm 0.005}$ | $-111.811_{\pm 0.002}$ | $-109.157_{\pm 0.000}$ |
| DIS-GMP | LV | $-5152.728_{\pm 18.004}$ | $-73.777_{\pm 0.007}$ | $-86.456_{\pm 0.557}$ | $0.722_{\pm 0.005}$ | $-111.844_{\pm 0.000}$ | $-109.443_{\pm 0.000}$ |

Table 7: Evidence lower bound (ELBO) values for various real-world benchmark problems, averaged across four seeds. Here, 'Div.' indicates if the model is trained using the Kullback-Leibler (KL) or log-variance (LV) divergence. The best overall results are highlighted in bold, with category-specific best results underlined. Blue and green shading indicate that the method uses learned Gaussian (GP) and Gaussian mixture priors (GMP), respectively.

sion models in Richter et al. (2023). The log-variance divergence is defined as

$$\mathcal{L}(\theta, \phi) = \mathbb{V}_{x_{0:N} \sim \mathcal{R}} \left[ \log \frac{\overleftarrow{p}^{\gamma, \phi}(x_{0:N})}{\overrightarrow{p}^{\theta, \phi}(x_{0:N})} \right], \tag{47}$$

where $\mathcal{R}$ describes a reference process, e.g. Equation (2a) where $u^\theta$ is replaced with an arbitrary control. In practice, one typically uses the generative process $\overrightarrow{p}^{\theta,\phi}$ with an additional stop gradient operator on the parameters (Richter et al., 2023). Not computing the expectations with respect to samples from the generative process significantly reduces memory consumption and does not require the prior distribution to be amendable to the reparameterization trick. The results are reported in Table 7 and follow the same experimental setting as outlined in the main part of the paper. We find that the KL divergence typically performs better than the log-variance divergence. Most significantly, the log-variance divergence seems to be numerically unstable for the *Credit* target.

**Comparison to long-run Sequential Monte Carlo** We additionally compare diffusion samplers with a learned Gaussian mixture prior to a Sequential Monte Carlo with a high number of discretization steps $N$. The results are shown in Table 8 and Table 9. While long-run SMC significantly

| METHOD | $N$ | CREDIT | SEEDS | CANCER | BROWNIAN | IONOSPHERE | SONAR |
|---|---|---|---|---|---|---|---|
| MCD-GMP | 128 | $-585.276_{\pm0.013}$ | $-73.461_{\pm0.004}$ | $-88.562_{\pm0.243}$ | $0.993_{\pm0.003}$ | $-111.827_{\pm0.007}$ | $-109.197_{\pm0.004}$ |
| CMCD-GMP | 128 | $-585.162_{\pm0.002}$ | $-73.429_{\pm0.002}$ | $-78.402_{\pm0.037}$ | $1.087_{\pm0.001}$ | $-111.682_{\pm0.000}$ | $-108.634_{\pm0.000}$ |
| DIS-GMP | 128 | $-585.223_{\pm0.006}$ | $-73.492_{\pm0.003}$ | $-84.061_{\pm2.117}$ | $0.885_{\pm0.005}$ | $-111.811_{\pm0.002}$ | $-109.157_{\pm0.000}$ |
| DBS-GMP | 128 | $\mathbf{-585.148_{\pm0.002}}$ | $\mathbf{-73.418_{\pm0.001}}$ | $\mathbf{-78.160_{\pm0.063}}$ | $\mathbf{1.118_{\pm0.002}}$ | $\mathbf{-111.657_{\pm0.002}}$ | $-108.548_{\pm0.000}$ |
| SMC | 128 | $-698.403_{\pm4.146}$ | $-74.699_{\pm0.100}$ | $-194.059_{\pm0.613}$ | $-1.874_{\pm0.622}$ | $-114.751_{\pm0.238}$ | $-111.355_{\pm1.177}$ |
| | 256 | $-708.185_{\pm14.225}$ | $-73.972_{\pm0.034}$ | $-140.757_{\pm7.041}$ | $-0.360_{\pm0.136}$ | $-113.110_{\pm0.046}$ | $-109.822_{\pm0.630}$ |
| | 512 | $-686.335_{\pm18.333}$ | $-73.667_{\pm0.015}$ | $-137.028_{\pm2.336}$ | $0.414_{\pm0.048}$ | $-112.353_{\pm0.036}$ | $-109.197_{\pm0.420}$ |
| | 1024 | $-690.011_{\pm12.879}$ | $-73.532_{\pm0.038}$ | $-128.809_{\pm6.046}$ | $0.786_{\pm0.047}$ | $-111.962_{\pm0.018}$ | $-108.291_{\pm0.325}$ |
| | 2048 | $-672.602_{\pm15.229}$ | $-73.496_{\pm0.017}$ | $-128.376_{\pm3.504}$ | $0.992_{\pm0.036}$ | $-111.785_{\pm0.022}$ | $\underline{\mathbf{-108.261_{\pm0.565}}}$ |
| | 4096 | $-665.973_{\pm19.849}$ | $-73.438_{\pm0.004}$ | $-121.950_{\pm3.315}$ | $1.088_{\pm0.029}$ | $-111.692_{\pm0.013}$ | $-108.736_{\pm0.227}$ |

Table 8: Evidence lower bound (ELBO) values for various real-world benchmark problems and different numbers of discretization steps $N$, averaged across four seeds. The best overall results are highlighted in bold, with category-specific best results underlined. green shading indicate that the method Gaussian mixture priors (GMP).

| | | FASHION ($d = 784$) | | | |
|---|---|---|---|---|---|
| METHOD | $N$ | ELBO $\uparrow$ | $\Delta \log \mathcal{Z} \downarrow$ | $\mathcal{W}_\gamma^2 \downarrow$ | EMC $\uparrow$ |
| DIS-GMP+IMR | 128 | $\mathbf{-62.482_{\pm2.752}}$ | $\mathbf{27.645_{\pm3.118}}$ | $\mathbf{513.776_{\pm13.936}}$ | $\mathbf{0.780_{\pm0.089}}$ |
| SMC | 128 | $-12181.932_{\pm134.611}$ | $11747.518_{\pm139.292}$ | $6696.287_{\pm250.4}$ | $0.026_{\pm0.027}$ |
| | 256 | $-10095.076_{\pm1076.723}$ | $9901.113_{\pm1078.916}$ | $6018.423_{\pm197.144}$ | $\underline{0.191_{\pm0.112}}$ |
| | 512 | $-9340.232_{\pm803.694}$ | $9254.499_{\pm804.027}$ | $5821.422_{\pm396.492}$ | $0.141_{\pm0.140}$ |
| | 1024 | $-9229.557_{\pm742.223}$ | $9190.558_{\pm742.656}$ | $5610.511_{\pm283.885}$ | $0.075_{\pm0.129}$ |
| | 2048 | $-8472.660_{\pm281.288}$ | $8454.062_{\pm281.475}$ | $5718.030_{\pm328.909}$ | $0.257_{\pm0.038}$ |
| | 4096 | $-8399.465_{\pm153.637}$ | $8390.302_{\pm153.707}$ | $5583.099_{\pm179.370}$ | $0.102_{\pm0.108}$ |

Table 9: Results for Fashion target, averaged across four seeds and reported across different numbers of discretization steps $N$. Evaluation criteria include evidence lower bound ELBO, importance-weighted errors for estimating the log-normalizing constant $\Delta \log \mathcal{Z}$, and Sinkhorn distance $\mathcal{W}_2^\gamma$ and entropic mode coverage EMC. The best overall results are highlighted in bold, with category-specific best results underlined. Arrows ($\uparrow$, $\downarrow$) indicate whether higher or lower values are preferable, respectively. Orange shading indicates that the method uses iterative model refinement (IMR).

| METHOD | FUNNEL | SEEDS | BROWNIAN | IONOSPHERE | SONAR |
|---|---|---|---|---|---|
| DIS | $2.993_{\pm0.042}$ | $4.688_{\pm0.055}$ | $6.266_{\pm0.329}$ | $4.394_{\pm0.066}$ | $4.840_{\pm0.031}$ |
| DIS-GMP | $\underline{1.898_{\pm0.002}}$ | $\underline{2.367_{\pm0.008}}$ | $\underline{2.445_{\pm0.004}}$ | $\underline{2.736_{\pm0.004}}$ | $\underline{3.861_{\pm0.036}}$ |

Table 10: Variability in the dynamics of the learned model via time-integrated spectral norm of the control $S \times 10^2$ (see Equation (45)) for various benchmark problems. Both DIS and DIS-GMP use $N = 128$ diffusion steps. Here, DIS-GMP uses $K = 10$ components. Lower values indicate lower variability in the dynamics of the learned model.

increases ELBO values, GMP-based diffusion sampler yield superior results in most experiments. Moreover, the results on the *Fashion* target indicate that more discretization steps yield better ELBO, $\Delta \log \mathcal{Z}$ and Sinkorn distances, but are not able to prevent mode collapse as indicated by the low EMC values.

**Ablation: Variation of dynamics** We additionally compare the variability in the dynamics of the learned model between DIS and DIS-GMP via time-integrated spectral norm of the control $S$ (see Equation (45)). The results are shown in Table 10 and show that DIS-GMP indeed has less variation in the dynamics. These findings are also in line with those in Figure 3 and Table 2 where DIS-GMP has significantly higher ELBO values compared to DIS without learned prior.

# E    LATTICE $\phi^4$ THEORY

We apply our method to simulate a statistical lattice field theory near and beyond the phase transition. This phase transition marks the progression of the lattice from disordered to semi-ordered and ultimately to a fully ordered state, where neighboring sites exhibit strong correlations in sign and magnitude.

We study the lattice $\phi^4$ theory in $D = 2$ spacetime dimensions (distinct from the problem's dimensionality as described below). The random variables in this setting are field configurations $\phi \in \mathbb{R}^{L \times L}$, where $L$ represents the lattice extent in space and time. The density of these configurations is defined as

$$\pi(\phi) = \frac{e^{-U(\phi)}}{Z},$$

where the potential $U(\phi)$ is given by:

$$U(\phi) = -2\kappa \sum_x \sum_\mu \phi_x \phi_{x+\mu} + (1 - 2\lambda) \sum_x \phi_x^2 + \lambda \sum_x \phi_x^4. \tag{48}$$

Here, the summation over $x$ runs over all lattice sites, and the summation over $\mu$ considers the neighbors of each site. The parameters $\lambda$ and $\kappa$ are referred to as the bare coupling constant and the hopping parameter, respectively. Following Nicoli et al. (2021), we set $\lambda = 0.022$, identifying the critical threshold of the theory (the transition from ordered to disordered states) at $\kappa \geq 0.3$. Near this threshold, sampling becomes increasingly challenging due to the multimodality of the density, with modes becoming more separated for larger values of $\kappa$.

We conduct experiments for $\kappa \in \{0.2, 0.3, 0.5\}$ across various problem dimensions $d = L \times L$. The methods compared include DIS, DIS-GP, and DIS-GMP, each with $N = 128$ diffusion steps, as well as a long-run SMC sampler with $N = 4096$. For all methods, the initial support is set to 5, approximately covering the target's support for all tested values of $\kappa$. The tuned parameters of the HMC kernel for the SMC sampler are detailed in Table 4, while additional parameter settings are provided in Appendix C.2. Note that DIS (and its extensions) do not undergo hyperparameter tuning due to their end-to-end learning framework.

To compare the different methods, we utilize the negative variational free energy of the system, defined as:

$$-\mathcal{F} = \frac{1}{L^2} \log \mathcal{Z} \geq \frac{1}{L^2} \text{ELBO}. \tag{49}$$

This bound follows from the inequality $\log \mathcal{Z} \geq \text{ELBO}$ as discussed in Section 3.2 and provides a means of comparison between sampling methods. However, as the ELBO (and thus $\mathcal{F}$) is not sensitive to mode collapse (Blessing et al., 2024), and since samples from the target distribution are unavailable, we also qualitatively assess the methods by visualizing the (normalized) histogram of the average magnetization $M(\phi) = \sum_x \phi_x$ across lattice configurations $\phi$.

Quantitative results are presented in Table 11, with qualitative findings illustrated in Figure 12. The results indicate that learning the prior (i.e. DIS-GP/DIS-GMP) significantly improves free energy estimates compared to DIS without a learned prior. Moreover, Figure 12 demonstrates that DIS-GMP avoids mode collapse while achieving comparable or better free energy estimates than both DIS-GP and the long-run SMC sampler in the majority of settings. While SMC captures multimodality at the phase transition ($\kappa = 0.3$), it struggles with the multimodality in the fully ordered phase ($\kappa = 0.5$). By contrast, DIS and DIS-GP are prone to mode collapse. Lastly, the performance of DIS degrades significantly with increasing problem dimension $d$, which is mitigated when using a learned Gaussian or Gaussian mixture prior.

| METHOD | $\kappa$ | $d = 16$ | $d = 64$ | $d = 100$ | $d = 144$ | $d = 196$ | $d = 256$ |
|---|---|---|---|---|---|---|---|
| DIS | | $0.6263_{\pm 0.0000}$ | $0.6186_{\pm 0.0009}$ | $0.6166_{\pm 0.0008}$ | $0.6145_{\pm 0.0014}$ | $0.5997_{\pm 0.0011}$ | $0.5749_{\pm 0.0013}$ |
| DIS-GP | | $0.6274_{\pm 0.0000}$ | $0.6219_{\pm 0.0001}$ | $0.6200_{\pm 0.0001}$ | $\mathbf{0.6175}_{\pm 0.0000}$ | $\mathbf{0.6158}_{\pm 0.0001}$ | $0.6113_{\pm 0.0015}$ |
| DIS-GMP + IMR | 0.2 | $\mathbf{0.6276}_{\pm 0.0000}$ | $\mathbf{0.6232}_{\pm 0.0004}$ | $\mathbf{0.6231}_{\pm 0.0002}$ | $0.6166_{\pm 0.0027}$ | $0.6139_{\pm 0.0006}$ | $\mathbf{0.6156}_{\pm 0.0005}$ |
| SMC | | $0.6167_{\pm 0.0022}$ | $0.6175_{\pm 0.0005}$ | $0.6186_{\pm 0.0003}$ | $0.6164_{\pm 0.0072}$ | $0.6087_{\pm 0.0029}$ | $0.6142_{\pm 0.0099}$ |
| DIS | | $1.0653_{\pm 0.0195}$ | $1.0277_{\pm 0.0004}$ | $1.0217_{\pm 0.0004}$ | $1.0040_{\pm 0.0024}$ | $0.9534_{\pm 0.0017}$ | $0.8905_{\pm 0.0014}$ |
| DIS-GP | | $1.0831_{\pm 0.0021}$ | $1.0459_{\pm 0.0001}$ | $1.0411_{\pm 0.0001}$ | $1.0372_{\pm 0.0011}$ | $1.0343_{\pm 0.0010}$ | $1.0319_{\pm 0.0004}$ |
| DIS-GMP + IMR | 0.3 | $\mathbf{1.0940}_{\pm 0.0000}$ | $1.0496_{\pm 0.0033}$ | $1.0461_{\pm 0.0001}$ | $1.0406_{\pm 0.0001}$ | $\mathbf{1.0380}_{\pm 0.0007}$ | $1.0347_{\pm 0.0006}$ |
| SMC | | $1.0848_{\pm 0.0013}$ | $\mathbf{1.0514}_{\pm 0.0007}$ | $\mathbf{1.0488}_{\pm 0.0002}$ | $\mathbf{1.0437}_{\pm 0.0052}$ | $1.0339_{\pm 0.0043}$ | $\mathbf{1.0389}_{\pm 0.0098}$ |
| DIS | | $12.2545_{\pm 0.0004}$ | $12.2097_{\pm 0.0023}$ | $9.2610_{\pm 2.9035}$ | $11.7179_{\pm 0.0799}$ | $9.4392_{\pm 1.9044}$ | $10.8546_{\pm 0.0716}$ |
| DIS-GP | | $12.2735_{\pm 0.0000}$ | $12.2715_{\pm 0.0000}$ | $12.2692_{\pm 0.0012}$ | $12.2670_{\pm 0.0010}$ | $10.1714_{\pm 2.0923}$ | $\mathbf{12.2629}_{\pm 0.0001}$ |
| DIS-GMP + IMR | 0.5 | $\mathbf{12.3167}_{\pm 0.0001}$ | $\mathbf{12.2806}_{\pm 0.0003}$ | $\mathbf{12.2761}_{\pm 0.0003}$ | $\mathbf{12.2730}_{\pm 0.0002}$ | $\mathbf{12.2722}_{\pm 0.0000}$ | $12.2588_{\pm 0.0001}$ |
| SMC | | $12.3049_{\pm 0.0025}$ | $12.2707_{\pm 0.0013}$ | $12.2679_{\pm 0.0027}$ | $12.2499_{\pm 0.0101}$ | $12.2416_{\pm 0.0040}$ | $12.2407_{\pm 0.0024}$ |

Table 11: Lower bound values for negative variational free energy $-\mathcal{F}$ as defined in (49) for the lattice $\phi^4$ theory problem with different values for the space-time extend $\sqrt{d} = L$ averaged across two seeds. The best (i.e. the highest) overall results are highlighted in bold for each configuration of the hopping parameter $\kappa$ and space-time extend.

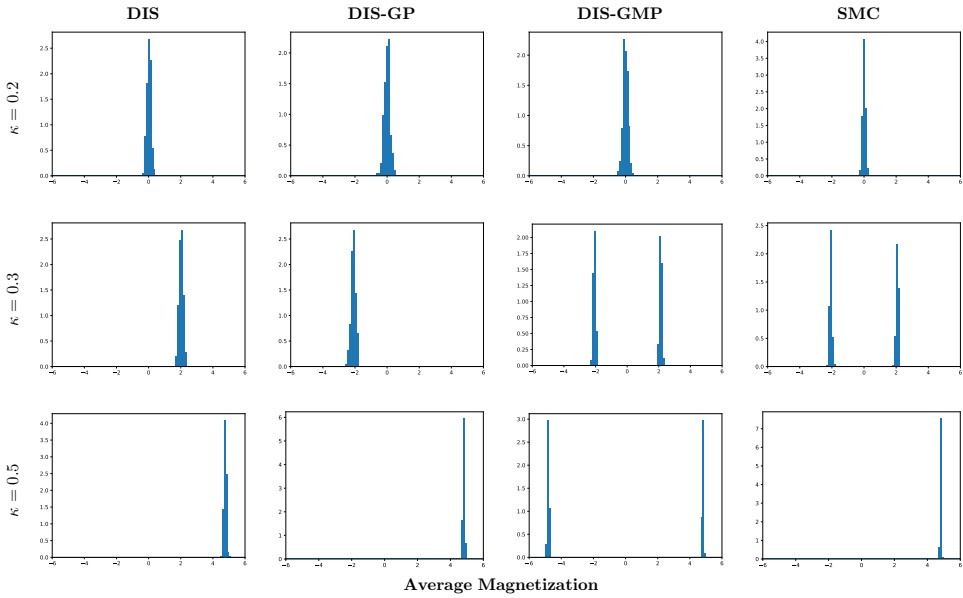

Figure 12: Normalized histogram of the average magnetization $M(\phi) = \sum_x \phi_x$ for 2000 samples $\phi \in \mathbb{R}^{L \times L}$ and space-time extend $L = 14$ for DIS, DIS-GP, DIS-GMP and long-run SMC for different values of the hopping parameter $\kappa$. The plots are generated using the same random seed 0.

