# OpenReview forum: "End-to-end Learning of Gaussian Mixture Priors for Diffusion Sampler"
_ICLR.cc/2025/Conference — ICLR 2025 Poster_

### Official Review · Reviewer_3sKi · 2024-10-29

**Soundness:** 4
**Presentation:** 3
**Contribution:** 3
**Rating:** 6
**Confidence:** 4

**Summary:**

In this paper, the authors propose an innovative sampling approach using the framework of controlled diffusions. They address current limitations in diffusion samplers, such as poor exploration, mode collapse, and numerical instabilities, which hinder model effectiveness. To counter these issues, they suggest parametrizing the prior of the backward diffusion process as a Gaussian Mixture Model (GMM). The authors detail how this adjustment can be applied to both denoising diffusions and Langevin diffusions. They introduce an iterative learning procedure that jointly optimizes the control networks and the GMM prior parameters. Through extensive experiments, the authors demonstrate that their approach outperforms traditional diffusion samplers across a wide range of tasks.

**Strengths:**

- The paper is well written
- The idea of parametrizing the prior seems novel

**Weaknesses:**

- Lack of multimodal experiments (while it was the main motivation according to Fig. 1)
- Competing algorithms seems to have been wrongly implemented as the control networks lack the $\nabla \log \pi$ information (according to L927-928)
- Sec 6 only considers metrics known to be unsuited for multimodal distributions (and especially to detect mode collapse C3) as per [1] (cited by the authors)
- The computational procedure is very heavy

**Questions:**

- Why not using the log-variance objective which as proven to be very effective ? [2] (Note that this objective wouldn't require the reparameterization trick on the prior)
- Could you provide an ablation study in a controlled environment (typically sampling Mixture of Gaussian) to actually challenge the procedure and assess that the claims were addressed ?
- How does your algorithm scale with the number of modes $K$ as GMM notoriously showcase numerical instabilities in this scenario ?
- Fig 4 and Fig 9 clearly show that the model mode collapsed as most of the generated images are almost identical. How do you explain this behavior ?
- According to Fig 6, the Funnel sampling procedure (for DIS) took over 5 hours. Is it actually competitive against MCMC samplers like MALA or HMC under the same computational budget ?
- It seems like SMC and CRAFT have been unfairly implemented. Indeed, as those algorithms use a different annealing scheme, there are no reasons to keep the same number of intermediate distributions. Moreover, as they are cheaper, this number should be increased. Could you re-run the experiments with increased budget ?
- Could you compare X-GMP against X-GP but adding the target informed parametrization of the controlled networks ? It seems like it could make a big difference.
- The problem of overparameterisation of the GMM (when the GMM as more mode than the target distribution) has two different interpretations between L479-483 and Fig 9. Could you clarify it ?
- Could you compute mode-collapse aware metrics such as the EUBO (and derivations) or the EMC from [1] ?

[1] Blessing, D., Jia, X., Esslinger, J., Vargas, F., & Neumann, G. (2024). Beyond ELBOs: A Large-Scale Evaluation of Variational Methods for Sampling. In Proceedings of the 41st International Conference on Machine Learning (pp. 4205–4229). PMLR.

[2] Nüsken, N., Richter, L. Solving high-dimensional Hamilton–Jacobi–Bellman PDEs using neural networks: perspectives from the theory of controlled diffusions and measures on path space. Partial Differ. Equ. Appl. 2, 48 (2021). https://doi.org/10.1007/s42985-021-00102-x

---

> ### Author Response · Authors · 2024-11-23
> **Response to Reviewer 3sKi**
>
> We thank the reviewer for taking the time to review our work and the many helpful comments and suggestions. We hope the following replies address the questions and concerns raised. Additionally, we uploaded a revised version of the manuscript that includes several changes as outlined in the sequel. The changes are highlighted using the color magenta.
>
> ---
>
> > Why not using the log-variance objective which has proven to be very effective? [2] (Note that this objective wouldn't require the reparameterization trick on the prior)
> >
>
> The log-variance loss is indeed a very interesting alternative to the KL divergence. However, we found the performance to be overall worse compared to KL and in some cases unstable. We included additional results in Appendix D, “Ablation: Kullback-Leibler vs. Log-Variance divergence”.
>
> Moreover, we thank the reviewer for the very interesting fact about the reparameterization trick. We mentioned in the main part of the paper that one can optimize arbitrary divergences and added a footnote that the reparameterization trick is only necessary if the optimization requires expectations with respect to samples from the generative process $\mathcal{P}^{\theta}$.
>
> ---
>
> > Could you provide an ablation study in a controlled environment (typically sampling Mixture of Gaussian) to actually challenge the procedure and assess that the claims were addressed?
> >
>
> We thank the reviewer for the great suggestion. We added an ablation study on a two-dimensional Gaussian mixture target in Appendix D, “Ablation on Gaussian Mixture Target”.
>
> ---
>
> > How does your algorithm scale with the number of modes $K$ as GMM notoriously showcase numerical instabilities in this scenario?
> >
>
> We found that end-to-end training of the GMM parameters works surprisingly well and didn't observe any numerical instabilities even when using $K>10$ components (see e.g. Figure 7 & 8).
>
> ---
>
> > Fig 4 and Fig 9 clearly show that the model mode collapsed as most of the generated images are almost identical. How do you explain this behavior?
> >
>
> DIS-GMP in Fig. 4 and 9 used a ‘naively’ initialized GMM with zero mean and unit variance components. Note that this is not well-suited for tackling challenging exploration problems such as high-dimensional targets with well-separated modes. For this reason, we additionally introduced the IMR strategy to facilitate mode discovery which yields better samples at the cost of additional design choices.
>
> ---
>
> > According to Fig 6, the Funnel sampling procedure (for DIS) took over 5 hours. Is it actually competitive against MCMC samplers like MALA or HMC under the same computational budget ?
> >
>
> MCMC algorithms (which are not model-based), such as MALA or HMC, are only guaranteed to converge to the target distribution $\pi$ in infinite time. In contrast, learned samplers such as DIS are guaranteed to sample from the target in a finite time horizon $T$ (given that the learned model has learned the optimal control).
>
> If time permits we will add another experiment that compares diffusion-based sampler to traditional MCMC methods under the same computational budget. However, we want to note that these methods are known to have problems mixing for complicated targets such as with the Funnel target or multimodal densities  [1].
>
> ---
>
> > It seems like SMC and CRAFT have been unfairly implemented. Indeed, as those algorithms use a different annealing scheme, there are no reasons to keep the same number of intermediate distributions. Moreover, as they are cheaper, this number should be increased. Could you re-run the experiments with increased budget ?
> >
>
> To the best of our knowledge, SMC, i.e. AIS with resampling uses the same annealing as MCD with the difference that MCD uses a learned control network to counteract the suboptimality of the backward Markov kernels used by SMC [2, Section 2.2 & 3]. Moreover, CRAFT converges to the SDE described by CMCD in the limit of $N \rightarrow \infty$ (see [3], Section 5.2). The major difference between CRAFT and CMCD is the parameterization for the transition densities, where CRAFT uses normalizing flows and CMCD a control network. In light of this, we do believe that the comparison is fair, in the sense, that all algorithms use the same number of discretization steps.
>
> Nevertheless, we included a comparison to a long-run SMC sampler which uses up to $N=4096$ discretization steps. Please note that using a high number of discretization steps is typically not possible for CRAFT since one has to maintain a normalizing flow model for every step in memory. We included the additional results in Appendix D, “Comparison to long-run Sequential Monte Carlo”.

---

> > ### Author Response · Authors · 2024-11-23
> > **Response to Reviewer 3sKi (cont.)**
> >
> > > Competing algorithms seems to have been wrongly implemented as the control networks lack the $\nabla \log \pi$   information (according to L927-928) […] Could you compare X-GMP against X-GP but adding the target informed parametrization of the controlled networks ?
> > >
> >
> > We thank the reviewer for this comment. We purposefully did not include the $\nabla \log \pi$  information in the network architecture for several reasons. First, MCD, CMCD, and DBS already contain the target gradient information as part of the drift of the SDE. For these methods, the authors of the previously mentioned works did also not use the target information as part of the model architecture. For DIS we found that a) using $\nabla \log \pi$   information led to similar results but often more unstable training curves and b) that it can be desirable not having to evaluate the gradient of the target in every iteration as this can be very costly and does not scale to targets where this evaluation is expensive.
> >
> > Nevertheless, we conducted an additional ablation study comparing the performance of DIS with and without the $\nabla \log \pi$   information. The results can be found in Appendix D “Ablation: Influence of the control architecture”.
> >
> > ---
> >
> > > The problem of overparameterization of the GMM (when the GMM as more mode than the target distribution) has two different interpretations between L479-483 and Fig 9. Could you clarify it?
> > >
> >
> > You are correct that Figure 9 shows a slight decrease in ELBO and $\Delta \log Z$ performance as $K$ increases, which is not entirely consistent with the results in Figure 5. However, this observation aligns with the results in Figure 4, where DIS-GMP demonstrates slightly worse ELBO and $\Delta \log Z$ values compared to DIS-GP. We attribute this discrepancy to two main reasons:
> >
> > 1. **Limitations of ELBO** and $\Delta \log Z$  **as Metrics for Multi-Modal Targets:** As you pointed out, for multi-modal targets with well-separated modes, ELBO and $\Delta \log Z$  values often fail to reflect sample quality accurately. In such cases, these metrics tend to favor models that fit a single mode perfectly. In contrast, the Sinkhorn distance, which leverages ground-truth samples, provides a more reliable criterion for evaluating sample quality. For the Sinkhorn distance, we observe consistent improvements as $K$ increases, which aligns with the qualitative results shown in Figures 4 and 9.
> > 2. **Increased Complexity of Control Functions for Larger** $K$**:** With higher $K$, the diffusion model has to learn more complex control functions, as it needs to operate over the support of the entire Gaussian Mixture Model (GMM) rather than a single Gaussian. This added complexity can introduce more opportunities for approximation errors, which may negatively impact ELBO and $\Delta \log Z$  values compared to using a single learnable Gaussian. Nevertheless, the resulting samples from DIS-GMP are closer to the target distribution in terms of optimal transport (as indicated by the Sinkhorn distance). Importantly, these errors remain significantly smaller than those observed with non-learnable priors, as demonstrated in Figure 4.
> >
> > We thank the reviewer again and agree that these results require further clarification. To address this, we have expanded the discussion of these results in Section 6.2 of the revised manuscript.
> >
> > ---
> >
> > > Sec 6 only considers metrics known to be unsuited for multimodal distributions (and especially to detect mode collapse C3) as per [1] (cited by the authors) […] Could you compute mode-collapse aware metrics such as the EUBO (and derivations) or the EMC from [1] ?
> > >
> >
> > We thank the reviewer for the suggestion. Although we consider the Sinkhorn distance to be a mode-collapse-aware metric, we additionally added the EMC metric to Table 4 (and included it in the new GMM experiment). We are currently having trouble achieving EUBO values comparable to those reported in [4]. We are currently investigating this issue.
> >
> > However, we already included additional information about the EUBO and EMC in Appendix C.4.
> >
> > ---
> >
> > > Lack of multimodal experiments
> > >
> >
> > In addition to the newly added GMM target Appendix D, we plan on adding the multimodal $\phi^4$ lattice field theory target [5]. If time permits, we include the results in a revised version and notify the reviewer.

---

> > > ### Author Response · Authors · 2024-11-23
> > > **Response to Reviewer 3sKi (cont.)**
> > >
> > > > The computational procedure is very heavy
> > > >
> > >
> > > While we agree that using Gaussian mixture priors come with an additional overhead caused by the necessity to evaluate the likelihood of the GMM in every step of the diffusion process, we want to stress that most diffusion-based methods additionally require evaluating the target density at every step. For experiments where it is expensive to evaluate the target, e.g. if the target is a big neural network or requires simulating molecular dynamics, the relative cost of evaluating the GMM likelihood becomes less pronounced or even negligible.
> > >
> > > The results reported in the paper are for DIS which only requires evaluating the GMM likelihood. We additionally included a comparison between the wallclock time for DIS and CMCD on the Fashion target in Table 5. In such a setting, it is easier to see that the relative cost added by the GMP is minor.
> > >
> > > ---
> > >
> > > Again, we would like to thank the reviewer for the many helpful questions that significantly improved the quality of our work. We welcome the opportunity to address any additional concerns or questions they may have.
> > >
> > > ---
> > > References
> > >
> > > [1] Radford M Neal. Slice sampling. The Annals of Statistics, 31(3):705–767, 2003.
> > >
> > > [2] Doucet, Arnaud, et al. "Score-based diffusion meets annealed importance sampling." *Advances in Neural Information Processing Systems* 35 (2022): 21482-21494.
> > >
> > > [3] Arbel, Michael, Alex Matthews, and Arnaud Doucet. "Annealed flow transport monte carlo." *International Conference on Machine Learning*. PMLR, 2021.
> > >
> > > [4] Nicoli, Kim A., et al. "Estimation of thermodynamic observables in lattice field theories with deep generative models." *Physical review letters* 126.3 (2021): 032001.
> > >
> > > [5] Blessing, D., Jia, X., Esslinger, J., Vargas, F., & Neumann, G. (2024). Beyond ELBOs: A Large-Scale Evaluation of Variational Methods for Sampling. In Proceedings of the 41st International Conference on Machine Learning (pp. 4205–4229). PMLR.

---

> > > > ### Comment · Reviewer_3sKi · 2024-11-23
> > > >
> > > > I want to deeply thank the authors for this great answer to my questions. I am increasing my score to 6 and I will be ready to increase it further depending on the incoming experiments as well as the discussion with other reviewers.

---

> > > > > ### Author Response · Authors · 2024-11-28
> > > > > **Reply to Reviewer 3sKi (cont.)**
> > > > >
> > > > > We sincerely thank the reviewer for their positive feedback and are pleased to hear that our clarifications were helpful in addressing the questions.
> > > > >
> > > > > Moreover, we are delighted to inform the reviewer that we were able to conduct additional experiments to strengthen our work further. Specifically, we investigated the lattice $\phi^4$ field theory problem where we analyzed three distinct stages of the lattice’s phase transition, two of which exhibit multimodal behavior. Furthermore, the problem was studied across varying dimensionalities, ranging from $d=16$ to $d=256$. These results have been added to Appendix E, “Lattice $\phi^4$ Theory”.

---

### Official Review · Reviewer_PAqT · 2024-11-03

**Soundness:** 3
**Presentation:** 3
**Contribution:** 3
**Rating:** 6
**Confidence:** 3

**Summary:**

This paper proposes an end-to-end learnable Gaussian mixture priors (GMPs) to address challenges in diffusion-based sampling methods including poor exploration, non-linear dynamics and mode collapse. Generally speaking, the paper contains some interesting ideas, and some experimental results are provided to demonstrate the performance improvement of the proposed method.

**Strengths:**

The paper presents an approach for improving diffusion-based sampling techniques by introducing end-to-end learnable Gaussian Mixture Priors (GMPs). The authors identify three key challenges in diffusion models: lack of exploration capabilities (C1), non-linear dynamics requiring many diffusion steps (C2), and mode collapse due to reverse KL minimization (C3). To address these challenges, the paper proposes GMPs, which offer improved control over exploration, adaptability to target support, and increased expressiveness to counteract mode collapse. Experimental results are provided to demonstrate performance improvement of the proposed methods.

**Weaknesses:**

The main concern of the paper is that the considered numerical experiments are oversimplified. The authors only considered logistic regression models and the fashion target on the MNIST fashion dataset in experiments. It is well known that diffusion models are widely used for generating samples with complicated priors, such as high-quality real-world images. Yet, from the numeral results provided in the paper, the reviewer cannot see any potential of the proposed method in this respect. As such, it is not convincing to claim that the proposed methodology can indeed solve the problems (C1)-(C3) of the existing diffusion models.

**Questions:**

1. How can the proposed iterative model refinement (IMR) strategy be further improved to optimize the selection criteria for adding new components and dynamically adjust the number of components during training?

2. As shown in the left side of Figure 4, the proposed IMR strategy seems not work well in all metrics (DIS-GMP, DIS-GMP-IMR), how could this be explained?

3. As introduced in Section 5, the IMR strategy requires to optimize parameters until a predefined criterion is met, how to choose the criterion and how does the criterion affect the training result?

4. The mean of a new added Gaussian component is chosen according to (22), how about the choice of variance?

5. What are the potential limitations or drawbacks of using Gaussian mixture priors compared to other types of learnable priors, and how could these be addressed?

---

> ### Author Response · Authors · 2024-11-23
> **Response to Reviewer PAqT**
>
> We thank the reviewer for taking the time to review our work and for the many helpful questions. We hope the following replies address the questions and concerns raised. Additionally, we uploaded a revised version of the manuscript that includes several changes as outlined in the sequel. All changes are highlighted using the color magenta.
>
> ---
>
> > The main concern of the paper is that the considered numerical experiments are oversimplified. […] The authors only considered logistic regression models and the fashion target on the MNIST fashion dataset in experiments.
> >
>
> We appreciate the reviewer’s concern. However, we would like to highlight that our work includes the majority of the target densities presented in a recent sampling benchmark study [1]. Moreover, different from generative modeling, sampling problems with $d=100$ are already considered medium to high dimensional. See e.g. [2, 3] for recent works that only consider problems with $d \leq 50$. As discussed in [1, Section 7.1], the $d=784$ dimensional Fashion MNIST target is recognized as a highly challenging problem, where alternative approaches struggle to achieve satisfactory sample quality and diversity.
>
> It is important to clarify that the Fashion target is only loosely related to the Fashion MNIST dataset. Specifically, the dataset is utilized solely for training a normalizing flow, which subsequently serves as the target density. The actual dataset itself is not employed for training the diffusion sampler. This somewhat artificial problem introduces several significant challenges, with the most notable being the presence of well-separated modes in a high-dimensional space.
>
> ---
>
> > It is well known that diffusion models are widely used for generating samples with complicated priors, such as high-quality real-world images.
> >
>
> We believe there may be a slight misunderstanding regarding the specific context of our paper. In *posterior sampling*, the target distribution can be expressed as $\pi(x) = p(x|y) = \frac{p(y|x)p(x)}{Z}$, where $y$ is a given measurement and $p(x)$ and $p(y|x)$ are the prior and likelihood, respectively. For certain problems, the prior can also be more complex, such as those encountered in inverse problems involving image, audio, or video distributions [4,5,6,7]. In settings that differ from ours, where samples from the prior $p(x)$ are available, recent methods leverage *diffusion priors*, i.e., diffusion models pre-trained, to facilitate sampling from $\pi$. This is, however, different from our (more general) setting where minimal assumptions about the specific form of the target are made. Moreover, the term `prior', different from Bayesian inference, is commonly used in diffusion-based sampling (see e.g. [3]) to refer to the starting distribution of the stochastic differential equation which transports samples to the target.
>
> ---
>
> > How can the proposed iterative model refinement (IMR) strategy be further improved to optimize the selection criteria for adding new components and dynamically adjust the number of components during training?
> >
>
> This is indeed an interesting question we already highlighted in the future work section. One interesting avenue for future research is the improvement of the generation of candidate samples. We believe that an interesting direction is to use Markov chains where the invariant distribution is the target on higher temperatures $T$ for $\pi(x,T) \propto \exp \frac{-E(x)}{T}$. This would yield biased samples but facilitates the mixing of the Markov chain and thus encourages exploration of different regions of the target support which can help to find promising candidates for initializing new means of the Gaussian mixture prior.
>
> ---
>
> > The mean of a newly added Gaussian component is chosen according to (22), how about the choice of variance?
> >
>
> We thank the reviewer for the question and apologize for missing this detail in the original submission. The Gaussian components were initialized using unit variance. We added the corresponding details in Appendix C.2 in the revised version.

---

> ### Author Response · Authors · 2024-11-23
> **Response to Reviewer PAqT (cont.)**
>
> > As shown in the left side of Figure 4, the proposed IMR strategy seems not work well in all metrics (DIS-GMP, DIS-GMP-IMR), how could this be explained?
> >
>
> We thank the reviewer for the question. We attribute the slightly worse ELBO and log Z values to the following two reasons:
>
> 1. **Limitations of ELBO** and $\Delta \log Z$  **as Metrics for Multi-Modal Targets:** For multi-modal targets with well-separated modes, ELBO and $\Delta \log Z$  values often fail to reflect sample quality accurately. In such cases, these metrics tend to favor models that fit a single mode perfectly which was also observed in [1]. In contrast, the Sinkhorn distance, which leverages ground-truth samples, provides a more reliable criterion for evaluating sample quality. For the Sinkhorn distance, we observe consistent improvements as $K$ increases, which aligns with the qualitative results shown in Figures 4 and 9.
> 2. **Increased Complexity of Control Functions for Larger** $K$**:** With higher $K$, the diffusion model has to learn more complex control functions, as it needs to operate over the support of the entire Gaussian Mixture Model (GMM) rather than a single Gaussian. This added complexity can introduce more opportunities for approximation errors, which may negatively impact ELBO and $\Delta \log Z$  values compared to using a single learnable Gaussian. Nevertheless, the resulting samples from DIS-GMP are closer to the target distribution in terms of optimal transport (as indicated by the Sinkhorn distance). Importantly, these errors remain significantly smaller than those observed with non-learnable priors, as demonstrated in Figure 4.
>
> We thank the reviewer again and agree that these results require further clarification. To address this, we have expanded the discussion of these results in Section 6.2 of the revised manuscript.
>
> ---
>
> > As introduced in Section 5, the IMR strategy requires to optimize parameters until a predefined criterion is met, how to choose the criterion and how does the criterion affect the training result?
> >
>
> We chose to use 500 training iterations as an ad-hoc decision, as it worked well in practice; through visual inspection of model samples, we observed that newly added components focused on different modes. Consequently, we kept this value without exploring alternative values. To provide further insight, we included an ablation study in Appendix D, “Ablation: Iterations for IMR”, where we find that the model performs robustly across various choices for the iteration at which new components are added. Nevertheless, the concept of iteratively adding components is important, such that when a new component is added, the initialization is informed by the already existing mixture model (note that the heuristic in Eq. 22 depends on the likelihood of the GMP) to prevent initializing components at the same location.
>
> ---
>
> > What are the potential limitations or drawbacks of using Gaussian mixture priors compared to other types of learnable priors, and how could these be addressed?
> >
>
> To the best of our knowledge, we are the first to introduce learnable priors in the context of diffusion-based sampling. As mentioned in an earlier response, Gaussian Mixture Priors (GMPs) offer notable flexibility but can also complicate the training of control functions. This complexity arises because the control functions must operate over the support of the entire Gaussian Mixture Model (GMM), rather than a single Gaussian, which can lead to approximation errors.
>
> We believe these errors could be significantly mitigated by employing more sophisticated architectures, as opposed to the two-layer MLPs used in our current work. Exploring more advanced architectures remains an exciting direction for future research.
>
> ---
>
> We thank the reviewer for their feedback and hope that our responses have successfully addressed the questions and concerns raised. Should the reviewer have any additional questions or further concerns, we would be more than happy to provide further clarifications or answers

---

> > ### Author Response · Authors · 2024-11-23
> > **Response to Reviewer PAqT (cont.)**
> >
> > References
> >
> > [1] Blessing, D., Jia, X., Esslinger, J., Vargas, F., & Neumann, G. (2024). Beyond ELBOs: A Large-Scale Evaluation of Variational Methods for Sampling. In Proceedings of the 41st International Conference on Machine Learning (pp. 4205–4229). PMLR.
> >
> > [2] Richter, L., & Berner, J. (2023). Improved sampling via learned diffusions. ICLR 2024.
> >
> > [3] Berner, J., Richter, L., & Ullrich, K. (2022). An optimal control perspective on diffusion-based generative modeling. TMLR.
> >
> > [4] Chung, H., Kim, J., Mccann, M. T., Klasky, M. L., & Ye, J. C. (2022). Diffusion posterior sampling for general noisy inverse problems. *arXiv preprint arXiv:2209.14687*.
> >
> > [5] Hyungjin Chung, Byeongsu Sim, Dohoon Ryu, and Jong Chul Ye. Improving diffusion models
> > for inverse problems using manifold constraints. Advances in Neural Information Processing
> > Systems, 35:25683–25696, 2022b
> >
> > [6] Jiaming Song, Arash Vahdat, Morteza Mardani, and Jan Kautz. Pseudoinverse-guided diffusion models for inverse problems. In International Conference on Learning Representations, 2022
> >
> > [7] Benjamin Boys, Mark Girolami, Jakiw Pidstrigach, Sebastian Reich, Alan Mosca, and O Deniz
> > Akyildiz. Tweedie moment projected diffusions for inverse problems. arXiv preprint
> > arXiv:2310.06721, 2023.

---

> > > ### Author Response · Authors · 2024-11-25
> > > **Feedback inquiry for reviewer PAqT**
> > >
> > > Dear Reviewer PAqT,
> > >
> > > As the period allowing for modifications on the document is approaching its end, we would like to kindly inquire whether we have adequately addressed your concerns. If there are any remaining issues, we would greatly appreciate it if you could share them with us, allowing us sufficient time to respond thoroughly.
> > >
> > > Thank you again for your time and valuable feedback!

---

> ### Comment · Reviewer_PAqT · 2024-11-27
>
> I sincerely thank the authors for the detailed reply. I still have a remaining concern regarding the performance comparison of the proposed method with the SOTA in the field. Particularly, if possible, please provide comparisons with score-based plug-and-play methods, such as [R1], [R2].
>
> [R1] Yang Song and Stefano Ermon. Generative modeling by estimating gradients of the data distribution. Advances in neural information processing systems, 32, 2019.
>
> [R2] Yang Song, Jascha Sohl-Dickstein, Diederik P Kingma, Abhishek Kumar, Stefano Ermon, and Ben Poole. Score-based generative modeling through stochastic differential equations. arXiv preprint arXiv:2011.13456, 2020.

---

> > ### Author Response · Authors · 2024-11-27
> > **Reply to Reviewer PAqT (cont.)**
> >
> > We sincerely thank the reviewer for their engagement in the discussion. We believe there may be a misunderstanding regarding the specific setting of our paper. The references the reviewer provided address scenarios where access to a dataset is assumed. In such cases, one can employ score- or bridge-matching losses to optimize the diffusion model. However, this is not applicable to the setting we consider, which is often referred to as the "sampling/VI" setting. In this setting, we only have access to an unnormalized density (or energy function) $\rho(x) = \pi(x) Z$ and not to a dataset.
> >
> > To clarify, the optimization objective in the works mentioned by the reviewer (e.g., R1 and R2) is based on the denoising score matching objective:
> >
> > $$
> > \text{argmin}_{\theta}  \mathbb{E}\_{t} \mathbb{E}\_{x\_0 \sim p\_{\text{data}}}
> > \mathbb{E}\_{x\_t \sim p\_{t|0}(\cdot|x\_0)}\left[\|s\_{\theta}(x(t),t) - \nabla\_{x}\log p\_{t|0}(x\_t|x\_0)\right],
> > $$
> >
> > where the expectation is computed with respect to $p_{\text{data}}$ requiring access to a dataset with samples $x_0 \sim p_{\text{data}}$.  The same applies to alternative matching objectives such as sliced score matching as discussed in R1.
> >
> > In contrast, the setting we address optimizes the reverse Kullback-Leibler (KL) divergence:
> >
> > $$
> > D_{\text{KL}}\left(p^{\theta}(x)\| \pi(x)\right) = {\mathbb{E}_{x \sim p^{\theta}} \left[     \log \frac{p^{\theta}(x)}{\rho(x)}    \right]} + \log Z.
> > $$
> >
> > which does not require samples from a dataset but instead uses samples from the model $p^{\theta}$. This formulation relies on access to an unnormalized density $\rho(x) = \pi(x) Z$.
> >
> > This distinction is crucial and was emphasized in Section 3.2 (L200-205):
> >
> > “Note that the VI setting for optimizing diffusion models is different from techniques used when samples from the target are available ff.”
> >
> > We also referenced R1 in this section to explicitly highlight the difference from our work. If the reviewer has further suggestions for clarifying this distinction, we would be more than happy to incorporate them.
> >
> > ---
> >
> > > I still have a remaining concern regarding the performance comparison of the proposed method with the SOTA in the field.
> > >
> >
> > We thank the reviewer for raising this concern. However, we would like to emphasize that we compare our approach to state-of-the-art methods as identified in a very recent benchmark paper published at ICML 2024 [1]. Notably, all of these methods were published quite recently:
> >
> > - **CRAFT** - ICML 2022
> > - **MCD** - NeurIPS 2022
> > - **FAB** - ICLR 2023
> > - **DIS** - TMLR 2024
> > - **CMCD** - ICLR 2024
> > - **DBS** - ICLR 2024
> >
> > We hope this addresses the reviewer’s remaining concerns and would be happy to provide further clarifications or additional information if needed.
> >
> > ---
> >
> > References
> >
> > [1] Blessing, D., Jia, X., Esslinger, J., Vargas, F., & Neumann, G. (2024). Beyond ELBOs: A Large-Scale Evaluation of Variational Methods for Sampling. In Proceedings of the 41st International Conference on Machine Learning (pp. 4205–4229). PMLR.

---

> > > ### Author Response · Authors · 2024-11-29
> > > **Feedback inquiry for reviewer PAqT (cont.)**
> > >
> > > Dear Reviewer PAqT,
> > >
> > > As the discussion period is nearing its conclusion, we would like to kindly inquire once more whether we have addressed your concerns. If there are any remaining issues, we would be more than happy to provide further clarifications. If not, we would greatly appreciate it if you could consider updating your score.

---

> > > > ### Comment · Reviewer_PAqT · 2024-11-30
> > > >
> > > > Many thanks for the clarification. I have no further concerns, and have raised the score accordingly.

---

### Official Review · Reviewer_d8iD · 2024-11-05

**Soundness:** 3
**Presentation:** 4
**Contribution:** 3
**Rating:** 8
**Confidence:** 2

**Summary:**

The paper proposes using learned Gaussian mixture priors in diffusion samplers instead of the standard fixed, unimodal Gaussian. The authors experimentally demonstrate the advantages of having a multi-modal prior; better exploration, fewer diffusion steps, and avoiding mode collapse. To effectively train the learned Gaussian mixture prior model, the authors introduce an iterative refinement approach that gradually adds new modes to the prior distribution.

**Strengths:**

- **Novelty**: The idea of learning a mixture prior in a diffusion process is underexplored and could have significant impact on future works. To the best of my knowledge, there have been few papers that explore a learned prior in diffusion models. Additionally, the idea of gradually adding complexity to the prior is very interesting.
- **Evaluation**: The authors provide adequate comparisons between the proposed method and existing approaches that demonstrate the overall improvements when using the proposed model. The results show that their method better captures the different modes of the distributions learned.
- **Presentation**: The writing is succinct and helps the reader understand the necessary preliminaries before diving into the details of the proposed method. Even with limited prior knowledge of the different diffusion samplers out there, the paper is easy to follow.

**Weaknesses:**

- Some more ablations could help with understanding the specific advantages of the proposed method better. One of the issues of the unimodal Gaussian prior that is mentioned is the increased complexity in the learned dynamics. In some examples of Fig. 5 the EES only marginally improves with more modes, making the number of diffusion steps the more impactful hyperparameter. This hints towards the dynamics not being simpler with more modes, as we would expect fewer diffusion steps to also yield good EES in those cases. A more in-depth experiment that demonstrates the differences in dynamics could help ground the overall argument for a mixture prior.

**Questions:**

- For the exploration and mode collapse issues discussed, is it not possible to transform the target data distribution such that its support is covered by a single unit Gaussian, given that you use the knowledge of the support of the true distribution when adding new modes? How much worse is the base DIS compared to the learned and learned+mixture variants under this experiment setting?

---

> ### Author Response · Authors · 2024-11-25
> **Response to Reviewer d8iD**
>
> We thank the reviewer for taking the time to review our work and for the helpful questions and suggestions. We uploaded a revised version of the manuscript that includes several changes outlined in the sequel. All changes are highlighted in magenta.
>
> ---
>
> > For the exploration and mode collapse issues discussed, is it not possible to transform the target data distribution such that its support is covered by a single unit Gaussian, given that you use the knowledge of the support of the true distribution when adding new modes? How much worse is the base DIS compared to the learned and learned+mixture variants under this experiment setting?
> >
>
> We completely agree with the reviewer that ensuring a fair comparison requires the base version of DIS to utilize the same information as DIS-GMP+IMR. For the Fashion target, we indeed transformed the base version of DIS such that the prior covers the support of the target by setting the variance of the initial distribution to the same value that was used for initializing the candidate samples (see Appendix C.2).
>
> We sincerely apologize for missing this detail in the initial submission and added further details in Section 6.2 (Iterative Model Refinement (IMR)) and Appendix C.2 (Gaussian Priors (GP) and Gaussian Mixture Priors (GMP)).
>
> ---
>
> > Some more ablations could help with understanding the specific advantages of the proposed method better. One of the issues of the unimodal Gaussian prior that is mentioned is the increased complexity in the learned dynamics. In some examples of Fig. 5 the EES only marginally improves with more modes, making the number of diffusion steps the more impactful hyperparameter. This hints towards the dynamics not being simpler with more modes, as we would expect fewer diffusion steps to also yield good EES in those cases. A more in-depth experiment that demonstrates the differences in dynamics could help ground the overall argument for a mixture prior.
> >
>
> We thank the reviewer for the comment. Fig. 5 only compares between models where the prior is learned, ($K=1$ uses a learned Gaussian prior), in which case the variation in dynamics is typically not as pronounced as for a non-learned prior, see e.g. Fig. 1, C2 vs. C3.
>
> We, therefore, added a more in-depth experiment where we quantify the variation of the dynamics over time using the spectral norm of the Jacobian of the learned forward control, i.e.,
>
> \begin{equation}
> S  =  \mathbb{E} \left[\int_{0}^T\|\frac{\partial \sigma u^{\theta}(x,t)}{\partial x}\|_2 d t\right],
> \end{equation}
>
> on various tasks to compare the dynamics between DIS and DIS-GMP. The results are included in Appendix D (Ablation: Variation of dynamics) which indicate that DIS-GMP indeed has less variation in the dynamics. These findings are also in line with those in Fig. 3 and Table 2 where DIS-GMP has significantly higher ELBO values compared to DIS without learned prior.
>
> Furthermore, we provide an additional experiment in Appendix D (Ablation on Gaussian Mixture Target) where we study a two-dimensional Gaussian mixture model in order to provide further insights and comparisons.
>
> ---
>
> We thank the reviewer for their feedback and hope that our responses have successfully addressed the questions and concerns raised. Should the reviewer have any additional questions or further concerns, we would be more than happy to provide further clarifications or answers.

---

### Official Review · Reviewer_JR3v · 2024-11-09

**Soundness:** 3
**Presentation:** 3
**Contribution:** 3
**Rating:** 6
**Confidence:** 3

**Summary:**

This paper introduces a novel approach to diffusion-based sampling that incorporates end-to-end learnable Gaussian Mixture Priors (GMP), offering a more expressive alternative to conventional Gaussian priors. The flexibility of GMP mitigates mode collapse, enhances exploration capabilities, and reduces the need for extensive diffusion steps by alleviating non-linear dynamics. The authors also propose an Iterative Model Refinement (IMR) strategy, progressively increasing model complexity to optimize performance on high-dimensional, multi-modal targets. Extensive experiments demonstrate that using GMP in place of Gaussian priors in existing diffusion-based sampling methods achieves improved sampling performance and outperforms previous methods across synthetic and real-world datasets.

**Strengths:**

- The paper is well-structured, first identifying the limitations of Gaussian priors in diffusion-based sampling and then logically progressing to propose end-to-end learnable Gaussian Mixture Priors (GMP) as a solution. Each step in this argument is clearly justified, supported by sufficient background information and helpful illustrations, making it easy for readers to follow.

- The experimental design effectively supports the theoretical claims, providing empirical evidence of the advantages offered by GMP over traditional Gaussian priors. Additionally, the comprehensive review of related work and background offers readers a useful overview of recent developments in this field.

- The proposed method enables end-to-end learning of the prior distribution, allowing existing diffusion-based sampling methods, which previously relied on fixed naive priors, to benefit from a more optimized starting prior. This adaptability enhances sampling performance and broadens the applicability of the approach across various diffusion-based sampling methods.

**Weaknesses:**

- Although GMP can achieve competitive performance with a single Gaussian prior using fewer steps, high-dimensional data often require more complex dynamics and an increased number of components to handle multiple modes effectively. As both the number of components $K$ and steps $N$ grow, the computational cost increases substantially, raising concerns about the scalability of this approach for even higher-dimensional applications in the future.
- As acknowledged in the paper, there is no clear criterion for determining the appropriate number of mixture components needed for optimal performance. Furthermore, IMR relies on heuristic rules, such as a fixed number of iterations, to add new components, introducing additional hyperparameter tuning requirements for model optimization.
- Some minor inconsistencies and certain claims in the paper are not fully supported by the experimental results, which would benefit from further clarification. For instance, additional explanation may be needed for the DIS results in Figure 4.

**Questions:**

- In Section 5, it seems that the positioning of challenges C1 and C2 may have been reversed. Specifically, in lines 308–310, reducing the complexity of dynamics to minimize the required diffusion steps seems more directly related to the issue identified in the Introduction as C2 (lines 51–53). Conversely, enhancing the exploration capability of diffusion-based sampling methods to cover diverse regions seems more relevant to addressing C1.
- While Figure 6 presents wall-clock time for relatively low-dimensional cases, it would be helpful to understand how computational demands scale in higher-dimensional scenarios. Could you provide an analysis of the computational cost associated with varying the number of mixture components $K$ and diffusion steps $N$ for a more formal comparison? (e.g. Big-O notation)
- How does estimated memory consumption vary with changes in $K$ and $N$?
- In Figure 4, DIS-GMP performs worse than DIS-GP across all metrics of sampling quality. This outcome seems contrary to what the paper suggests; how might this be explained? Furthermore, while DIS-GMP + IMR with progressively learned components shows better Sinkhorn distance and qualitative sampling results, its ELBO and $\triangle$ log Z performance is lower than that of DIS-GP and DIS-GMP. This observation suggests that ELBO may have limitations in evaluating sampling performance for multi-modal target distributions (as also seen in Figure 9).
- Table 2 presents only ELBO results; does the Sinkhorn distance follow a similar trend to what was observed in the Funnel case?
- The IMR approach uses a criterion of adding components every 500 training iterations. Are there any ablation study results regarding this criterion for IMR?

Minor:
- In Equation 31 of Appendix A.1, the term $(-\frac{2}{\sigma^2})^2$ seems to have a sign inconsistency with Equation 32, where the third term should perhaps be positive.

---

> ### Author Response · Authors · 2024-11-25
> **Response to Reviewer JR3v**
>
> We thank the reviewer for taking the time to review our work and for the many helpful comments and suggestions. We hope the following replies address the questions and concerns raised. Additionally, we uploaded a revised version of the manuscript that includes several changes outlined in the sequel. All changes are highlighted in magenta.
>
> ---
>
> > In Section 5, it seems that the positioning of challenges C1 and C2 may have been reversed.
> >
>
> We thank the reviewer for bringing this to our attention. In the revised version, we swapped the positions accordingly.
>
> ---
>
> > While Figure 6 presents wall-clock time for relatively low-dimensional cases, it would be helpful to understand how computational demands scale in higher-dimensional scenarios. Could you provide an analysis of the computational cost associated with varying the number of mixture components  $K$ and diffusion steps $N$ for a more formal comparison? (e.g. Big-O notation)
> >
>
> Formally, we have the time-complexity $\mathcal{O}(NK)$. Here, the $K$ stems from the additional requirement of computing the likelihood of the GMP in every diffusion step. However, in practice, the evaluation of the likelihood for the GMP can be parallelized across its components, which substantially reduces the computational overhead. Moreover, most diffusion-based methods additionally require evaluating the target density at every step. For experiments where it is expensive to evaluate the target, e.g. if the target is a big neural network or requires simulating molecular dynamics, the relative cost of evaluating the GMM likelihood becomes less pronounced or even negligible.
>
> The results reported in the paper are for DIS which only requires evaluating the GMM likelihood and not the target. We additionally included a comparison between the wallclock time for DIS and CMCD on the high-dimensional Fashion target in Table 5. For CMCD, which requires target evaluations in every diffusion step, the relative cost added by the GMP is minor.
>
> Moreover, we added additional discussions in Appendix B (”Time complexity”) and Appendix D (”Wallclock time”).
>
> ---
>
> > How does estimated memory consumption vary with changes in $K$ and $N$?
> >
>
> Using the standard ‘discrete-then-optimize’ approach for optimizing the KL divergence in Eq. 11, which requires differentiation through the SDE, the memory consumption is linear in $K$ and $N$. In contrast, using e.g. the stochastic adjoint method for KL optimization [1], the memory consumption is constant. We used the former approach for our experiments due to its simplicity. However, if one uses a large number of components / diffusion steps, the latter option might be the preferred choice.
> On a sidenote, it is also possible to obtain constant memory consumption by using other loss functions such as the log-variance loss [2] or moment-loss [3].
>
> We thank the reviewer for the question. We included an additional discussion on memory consumption in Appendix B (”Memory consumption.”)
>
> ---
>
> > In Figure 4, DIS-GMP performs worse than DIS-GP across all metrics of sampling quality. This outcome seems contrary to what the paper suggests; how might this be explained?
> >
>
> We thank the reviewer for the question. We attribute the slightly worse ELBO and log Z values to the following two reasons:
>
> 1. **Limitations of ELBO** and $\Delta \log Z$  **as Metrics for Multi-Modal Targets:** For multi-modal targets with well-separated modes, ELBO and $\Delta \log Z$  values often fail to reflect sample quality accurately which was also observed in [4]. In such cases, these metrics tend to favor models that fit a single mode perfectly. In contrast, the Sinkhorn distance, which leverages ground-truth samples, provides a more reliable criterion for evaluating sample quality. For the Sinkhorn distance, we observe consistent improvements as $K$ increases, which aligns with the qualitative results shown in Figures 4 and 9.
> 2. **Increased Complexity of Control Functions for Larger** $K$**:** With higher $K$, the diffusion model has to learn more complex control functions, as it needs to operate over the support of the entire Gaussian Mixture Model (GMM) rather than a single Gaussian. This added complexity can introduce more opportunities for approximation errors, which may negatively impact ELBO and $\Delta \log Z$  values compared to using a single learnable Gaussian. Nevertheless, the resulting samples from DIS-GMP are closer to the target distribution in terms of optimal transport (as indicated by the Sinkhorn distance). Importantly, these errors remain significantly smaller than those observed with non-learnable priors, as demonstrated in Figure 4.
>
> We thank the reviewer again and agree that these results require further clarification. To address this, we have expanded the discussion of these results in Section 6.2 of the revised manuscript.

---

> > ### Author Response · Authors · 2024-11-25
> > **Response to Reviewer JR3v (cont.)**
> >
> > > Furthermore, while DIS-GMP + IMR with progressively learned components shows better Sinkhorn distance and qualitative sampling results, its ELBO and  log Z performance is lower than that of DIS-GP and DIS-GMP. This observation suggests that ELBO may have limitations in evaluating sampling performance for multi-modal target distributions (as also seen in Figure 9).
> > >
> >
> > Indeed, ELBO and $\log Z$ are limited in evaluating sample performance (see also answer above). This was explained and empirically shown in [4]. As suggested by Reviewer 3sKi, we added an additional evaluation metric in Table 4, tailored for evaluating the mode-coverage for multi-modal targets, namely the entropic mode coverage (EMC) which was introduced in [4].
> >
> > ---
> >
> > > Table 2 presents only ELBO results; does the Sinkhorn distance follow a similar trend to what was observed in the Funnel case?
> > >
> >
> > Computing the Sinkhorn distance requires samples from the target density $\pi$ which are not available for the 'real-world' tasks presented in Table 2. Consequently, for these tasks, we are limited to reporting ELBO and ESS values. We added further details Section 6.1 and Appendix C.4 to make this requirement more explicit.
> >
> > ---
> >
> > > The IMR approach uses a criterion of adding components every 500 training iterations. Are there any ablation study results regarding this criterion for IMR?
> > >
> >
> > We thank the reviewer for their question. The value ‘500’ was an ad-hoc decision that we kept without exploring alternative values. However, we agree that this design-choice requires further explanation. In response, we included an ablation study in Appendix D, “Ablation: Iterations for IMR”, where we find that the model performs robustly across various choices for the iteration at which new components are added. Nevertheless, the concept of iteratively adding components is important, such that when a new component is added, the initialization is informed by the already existing mixture model (note that the heuristic in Eq. 22 depends on the likelihood of the GMP) to prevent initializing components at the same location.
> >
> > ---
> >
> > Once again, we sincerely thank the reviewer for their many insightful questions, which have greatly contributed to improving the quality of our work. We welcome any further concerns or questions they may have and would be glad to address them.
> >
> > ---
> >
> > References
> >
> > [1] Li, X., Wong, T. K. L., Chen, R. T., & Duvenaud, D. (2020, June). Scalable gradients for stochastic differential equations. In *International Conference on Artificial Intelligence and Statistics* (pp. 3870-3882). PMLR.
> >
> > [2] Richter, L., & Berner, J. (2023). Improved sampling via learned diffusions. ICLR 2024
> >
> > [3] Hartmann, C., Kebiri, O., Neureither, L., & Richter, L. (2019). Variational approach to rare event simulation using least-squares regression. *Chaos: An Interdisciplinary Journal of Nonlinear Science*, *29*(6).
> >
> > [4] Blessing, D., Jia, X., Esslinger, J., Vargas, F., & Neumann, G. (2024). Beyond ELBOs: A Large-Scale Evaluation of Variational Methods for Sampling. In Proceedings of the 41st International Conference on Machine Learning (pp. 4205–4229). PMLR.

---

> ### Comment · Reviewer_JR3v · 2024-12-03
>
> Thank you for providing additional experiments and explanations in response to my questions. I sincerely appreciate the significant improvements made to the paper during the rebuttal period, as well as the effort you have put into enhancing the experimental results and explanations.
>
> However, I still have a few unresolved concerns:
> > **Limitations of $\text{ELBO}$ and $\Delta\text{ log }Z$ as Metrics for Multi-Modal Targets**: For multi-modal targets with well-separated modes, $\text{ELBO}$ and $\Delta\text{ log }Z$ values often fail to reflect sample quality accurately which was also observed in [4].
>
> > Computing the Sinkhorn distance requires samples from the target density
>  which are not available for the 'real-world' tasks presented in Table 2. Consequently, for these tasks, we are limited to reporting ELBO and ESS values. We added further details Section 6.1 and Appendix C.4 to make this requirement more explicit.
>
> The authors discuss the limitations of $\text{ELBO}$ and $\Delta\text{ log }Z$ as metrics for multi-modal targets. For real-world datasets, where the true density is unknown, $\text{ELBO}$ and $\text{ESS}$ are used as the primary metrics in this paper. Specifically, in the synthetic density case of FMNIST, where $\text{ELBO}$ has known limitations, the authors demonstrated the effectiveness of GMP and IMR using Sinkhorn distance or $\text{EMC}$ metrics.
>
> However, this raises the question: does the superior $\text{ELBO}$ performance shown in Table 2 for real-world datasets necessarily guarantee the learning of better posterior distributions, or is it only ensured for lower-dimensional cases?Demonstrating that the DIS-GMP + IMR model consistently achieves the highest $\text{ESS}$ performance even in cases where $\text{ELBO}$  and Sinkhorn distance yield conflicting results—such as with Fashion data—could help address this concern indirectly.
>
> > We thank the reviewer for their question. The value ‘500’ was an ad-hoc decision that we kept without exploring alternative values. However, we agree that this design-choice requires further explanation. In response, we included an ablation study in Appendix D, “Ablation: Iterations for IMR”, where we find that the model performs robustly across various choices for the iteration at which new components are added. Nevertheless, the concept of iteratively adding components is important, such that when a new component is added, the initialization is informed by the already existing mixture model (note that the heuristic in Eq. 22 depends on the likelihood of the GMP) to prevent initializing components at the same location.
>
> In Table 10, Although authors have provided results showing that performance improves as the number of IMR training steps increases, this insight seems to suggest that "more training within available resources and time is better" rather than offering a clear criterion for selecting the appropriate number of iterations. Introducing a less heuristic stopping criterion or conditions to determine the optimal IMR iterations could yield more robust follow-up results and insights.
>
> Based on the results so far, I keep my current score but remain open to further insights or discussions on these points.
>
> P.S. In Appendix A (Proofs), the terms in Eq. 33 are not eliminated to zero. Specifically, $\[-\frac{2}{\sigma^2}\]$ should be $\[\frac{2}{\sigma^2}\]$ in Eq. 32.

---

> > ### Author Response · Authors · 2024-12-03
> > **Reply to Reviewer JR3v (cont.)**
> >
> > We sincerely thank the reviewer for acknowledging the effort we put into improving the paper during the rebuttal phase. Furthermore, we thank the reviewer for engaging into the discussion.
> >
> > ---
> >
> > > However, this raises the question: does the superior ELBO performance shown in Table 2 for real-world datasets necessarily guarantee the learning of better posterior distributions, or is it only ensured for lower-dimensional cases?
> > >
> >
> > We thank the reviewer for the question and would like to elaborate further on our previous response. ELBO values indeed reflect the quality of the posterior distributions, as they are equivalent to the reverse KL divergence (up to a normalization constant and sign flip), which measures the closeness between the learned model and the true target distribution. Consequently, higher ELBO values indicate that the learned model is "closer" to the true target distribution. However, there are two notable drawbacks associated with this notion of closeness:
> >
> > 1. **Local Evaluation:** ELBO values are computed using samples from the model, which means they only evaluate the distribution within the model’s support. As such, they provide a "local" perspective and may not account for discrepancies in regions of the target density that the model fails to cover.
> > 2. **Mode-Seeking Behavior:** The reverse KL divergence inherently favors mode-seeking. It heavily penalizes samples in low-probability regions of the target while being more lenient about missing some modes entirely. In multimodal settings with well-separated modes, ELBO values are typically lower for models that fit a single mode perfectly.
> >
> > For these reasons, while ELBO values do reflect model performance, they have limitations when it comes to evaluating sample diversity for target distributions with well-separated modes.
> >
> > For real-world problems, with unknown support and the lack of samples from the target, ELBO and ESS values remain, to the best of our knowledge, the only viable performance metrics for assessing the model performance.
> >
> > ---
> >
> > > Demonstrating that the DIS-GMP + IMR model consistently achieves the highest ESS performance even in cases where ELBO and Sinkhorn distance yield conflicting results—such as with Fashion data—could help address this concern indirectly.
> > >
> >
> > We did not report ESS values for Fashion as they are mostly zero for such high dimensional tasks. Like ELBO, ESS is computed from density ratios evaluated on samples from the model, which introduces similar limitations as discussed in the previous answer. Consequently, ESS values are typically correlated with ELBO values.
> >
> > To elaborate further, the ESS reported in our work is sometimes referred to in the literature as *reverse ESS* because it is computed using samples from the model. An alternative metric is *forward ESS [1]*, which is computed using samples from the target density. However, for the real world tasks these samples are not available.
> >
> > ---
> >
> > > Introducing a less heuristic stopping criterion or conditions to determine the optimal IMR iterations could yield more robust follow-up results and insights.
> > >
> >
> > We completely agree with the reviewer that having a more principled criterion for determining when to add more components would be highly desirable.
> >
> > One intuitive approach we explored involves adding components when the optimization has converged. This could be formalized by monitoring the change in the (extended) ELBO, $\mathcal{L}$, and stopping if the absolute difference between consecutive iterations is smaller than a threshold, $\epsilon$, i.e.,
> >
> > $$
> > |\mathcal{L}(\theta^{(i)},\gamma^{(i)})-\mathcal{L}(\theta^{(i-1)},\gamma^{(i-1)})| \leq \epsilon,
> > $$
> >
> > where $i$ denotes the iteration. While this criterion is arguably more principled, it introduces a new design choice: the selection of $\epsilon$. Setting this threshold is not straightforward because it depends on the range of values that $\mathcal{L}$ can take. Since the optimal value of $\mathcal{L}$ corresponds to the true value of $\log Z$—a quantity that is typically unknown and problem-dependent—choosing a universally appropriate $\epsilon$ is challenging.
> >
> > To avoid this ambiguity, we opted for the simpler approach of using a fixed number of iterations as the criterion. While this is less principled, it is easy to implement and more intuitive.
> >
> > ---
> >
> > We hope that these answers have provided clarification, especially given that the discussion period is nearing its conclusion. We thank the reviewer again for their time and thoughtful feedback!
> >
> > ---
> >
> > References
> >
> > [1] Midgley, L. I., Stimper, V., Simm, G. N., Schölkopf, B., & Hernández-Lobato, J. M. (2022). Flow annealed importance sampling bootstrap. ICLR 2023

---

> > > ### Comment · Reviewer_JR3v · 2024-12-03
> > >
> > > Thank you for providing more detailed explanations regarding the unresolved questions.
> > >
> > > Your further responses were very helpful in deepening my understanding of the current situation presented in the paper.
> > >
> > > It would be great if potential solutions to the identified limitations could be explored in future works.
> > >
> > > I sincerely appreciate your hard work.

---

### Author Response · Authors · 2024-12-04
**General response**

We would like to extend our sincere gratitude to all the reviewers for their time, effort, and valuable feedback during the review process. We are glad that all reviewers have expressed their support for acceptance. Below, we provide a brief summary of the enhancements and additional studies included during the rebuttal phase in response to the reviewers' insightful comments:

- Two-dimensional Gaussian mixture target (GMM): We introduced an additional GMM target where we performed several analyses to empirically substantiate the claims made in the paper.
- Lattice $\phi^4$ theory: Moreover, we applied our method to simulate a statistical lattice field theory near, at, and beyond the phase transition at various problem dimensionalities. We provided qualitative and quantitative results that show that Gaussian mixture priors improve free energy estimates and counteract mode collapse.
- Comparison with a long-run Sequential Monte Carlo (SMC) sampler: We additionally compare diffusion samplers with a learned Gaussian mixture prior to SMC with a high number of discretization steps (up to 4096). We find that the diffusion sampler are on par or outperform SMC in most cases.
- Kullback-Leibler vs. Log-Variance divergence: We further compare using the KL divergence to using the log-variance divergence for optimizing the control functions. We find that the KL divergence typically performs better than the log-variance divergence.
- Influence of the control architecture: We further evaluate the performance using an
architecture that additionally incorporates the score of the target. Our results indicate that including the target score often leads to slightly worse results.
- Iterations for IMR: We additionally conducted an ablation study which considers different numbers of iterations for the iterative model refinement scheme at which new components are added. The results indicate that the performance remains stable for different choices of the hyperparameter.
- Variation of dynamics: We additionally compared the variability in the dynamics of the
diffusion models via the time-integrated spectral norm of the learned control function. The results show that Gaussian mixture priors indeed simplify the underlying dynamics compared to using a fixed Gaussian prior distribution.
- Wallclock-time: We provided an analysis showing that the relative cost of learned Gaussian mixture priors in terms of wallclock time is minimal, particularly in cases where the diffusion sampler necessitates the evaluation of the target density at each diffusion step.

Aside from the experiments and ablation studies, we extended the result discussion and included an additional performance criterion, the entropic mode coverage, which is well-suited for quantifying mode collapse.

---

### Meta-Review · Area_Chair_tJVw · 2024-12-22

**Metareview:**

This paper proposes a Gaussian mixture prior-based diffusion model for sampling from unknown distributions. The mixture modeling could alleviate the mode-collapse issue in learning the samplers. The effectiveness of the proposed method is shown on both synthetic and real-world data. The reviewers had concerns on the experiments and the complexity of the proposed method and these concerns were addressed in the rebuttal and discussion. The final rating of all reviewers are positive. I agree with the reviewers and recommend acceptance of this paper.

**Additional Comments On Reviewer Discussion:**

One reviewer had concerns on the limited datasets used in the experiments. However, this was because the reviewer had some misunderstanding of the generative modeling and sampling problems. The authors convinced the reviewer to raise the score. Also, there were some concerns on the complexity due to the mixture models, the authors also provided detailed responses to resolve this concern. All the other concerns were addressed after the rebuttal and discussion.

---

### Decision · Program_Chairs · 2025-01-22

Accept (Poster)